# Projected decrease in wintertime bearing capacity on different forest and soil types in Finland under a warming climate

Ilari Lehtonen[1], Ari Venäläinen[1], Matti Kämäräinen[1], Antti Asikainen[2], Juha Laitila[2], Perttu Anttila[2] and Heli Peltola[3]

[1]Finnish Meteorological Institute, 00101 Helsinki, Finland
[2]Natural Resources Institute Finland, 80100 Joensuu, Finland
[3]School of Forest Sciences, University of Eastern Finland, 80101 Joensuu, Finland

*Correspondence to*: Ilari Lehtonen (ilari.lehtonen@fmi.fi)

**Abstract.** Trafficability in forest terrain is controlled by ground-bearing capacity which is crucial from the timber harvesting point of view. In winter, soil frost affects the most the bearing capacity, and especially on peatland soils which have in general low bearing capacity. Ground frost affects similarly the bearing capacity of forest truck roads. Already 20 cm thick layer of frozen soil or 40 cm thick layer of snow on the ground may be sufficient for heavy forest harvesters. In this work, we studied the impacts of climate change on soil frost conditions, and consequently on ground-bearing capacity from the timber harvesting point of view. The number of days with good wintertime bearing capacity was modelled by using a soil temperature model with a snow accumulation model and wide set of downscaled climate model data until the end of the 21st century. The model was calibrated for different forest and soil types. The results show that by the mid-21st century, the conditions with good bearing capacity will decrease in wintertime in Finland most likely by about one month. The decrease in soil frost and wintertime bearing capacity will be more pronounced during the latter half of the century when drained peatlands may virtually lack soil frost in most of winters in southern and western Finland. The projected decrease in the bearing capacity, accompanied with increasing demand for wood harvesting from drained peatlands, induces a clear need for the development of sustainable and resource-efficient logging practices for drained peatlands. This is also needed to avoid unnecessary harvesting damages, like rut formation on soils and damage to tree roots and stems.

## 1 Introduction

Forests are the most important natural resource in Finland (Finnish Forest Research Institute, 2011). In 2016 the annual harvested volume of round wood in the country reached a new national record of 70 mill. m$^3$ (Natural Resources Institute Finland, 2017a, 2017b). There exists a pressure to increase this volume up to 80 mill. m$^3$ within the next couple of decades to meet the increasing wood demand of growing bioeconomy sector (Ministry of Employment and the Economy et al., 2014; Asikainen et al., 2016). Preferably the wood harvesting should be increased throughout the year to ensure continuous supply of raw material for wood using industry. This may be challenging due to differences in bearing capacity of forest soils with varying soil types and weather conditions.

Traditionally, in Finland logging has been mainly conducted during winter months. Still nowadays, approximately 60% of logging is carried out while the soil is frozen (Finnish Forest Research Institute, 2014). This is because the bearing capacity of forest sites is clearly higher during frozen than unfrozen soil conditions. Already 20 cm thick layer of frozen soil or 40 cm thick layer of snow can bear standard machines used in forest harvesting that weigh 15–30 tonnes (Eeronheimo, 1991; Kokkila, 2013). Small forest truck roads having light foundations do not either bear heavy timber trucks in wet road sections unless the soil is frozen (Kaakkurivaara et al., 2015). Multiple passes of a harvester and a loaded forwarder may cause ruts on the forest floor (Suvinen, 2006; Sirén et al., 2013; Pohjankukka et al., 2016). Operations in poorly bearing conditions increase this rut formation and damage caused to tree roots and stems as well as time and fuel consumption in the harvesting (Sirén et al., 2013; Pohjankukka et al., 2016). Furthermore, the condition of road network affects to the fuel consumption in timber transportation (Svenson and Fjeld, 2016).

More than half of the original peat bog area in Finland was drained for forestry mainly during the 1960s and 1970s (Simola et al., 2012). Consequently, peatlands consist nowadays one third of the Finnish forestry area and one fourth of the growing stock volume (Ala-Ilomäki et al., 2011). In increasing the wood harvesting, more intensive utilization of drained peatland forests has the largest potential (Ala-Ilomäki et al., 2011), because of a pronounced reduction of suitable logging sites on upland (mineral) soils (Uusitalo and Ala-Ilomäki, 2013). However, more intensive utilization of peatlands is a controversial issue. Peatlands representing sensitive forest sites are generally characterized by the most difficult forest harvesting conditions (Nugent et al., 2003; Uusitalo and Ala-Ilomäki, 2013; Uusitalo et al., 2015a). Moreover, in addition to the increasing demand of wood harvesting from drained peatlands, there exists a pressure to restore drained peatlands to natural state in order to maintain biodiversity and prevent carbon loss and nitrous oxide emissions from peatlands (Komulainen et al., 1999; Carroll et al., 2011; Pitkänen et al., 2013; Pärn et al., 2018).

The difficult harvesting conditions in drained peatlands are because of their inherently low ground-bearing capacity. Thus, logging is there generally conducted during winter when the soil is frozen (Ala-Ilomäki et al., 2011). Nevertheless, soil frost periods are on drained peatlands shorter than on upland forest sites because of the insulating effect of peat compared to upland (mineral) soils. In addition, ditch network forms obstacles for vehicles in peatlands. They are neither typically located next to the forest truck roads and trees are characterized by small size, uneven distribution and superficial roots (Laitila et al., 2013). Hence, wood harvesting on drained peatlands is in general less cost efficient than in upland forest sites (Ala-Ilomäki et al., 2011). Determined efforts are thus required to prolong the wood harvesting season from drained peatlands. This would provide an opportunity to increase the annual harvesting volume and confine seasonal variations in harvesting.

During the forthcoming decades, climate has been projected to become warmer due to the anthropogenic climate change (Collins et al., 2013; Knutti and Sedláček, 2013). The climate warming is expected to be pronounced on high latitudes like in Finland (Räisänen and Ylhäisi, 2015; Ruosteenoja et al., 2016). Previous studies have indicated that the climate warming leads unsurprisingly to reduced soil frost depth and shorter soil frost periods (e.g., Venäläinen et al., 2001a, 2001b; Kellomäki et al., 2010; Gregow et al., 2011; Jungqvist et al., 2014). This may shorten the winter harvesting season with good ground-bearing capacity, particularly on drained peatlands, having thus mainly negative impact on the forestry sector. Thus,

comprehensive understanding of expected changes in soil frost conditions is utmost important as these changes affect wood harvesting conditions and transport availability. This is also needed to develop logging practices that are at the same time both sustainable and cost-efficient and meet the required increase in wood supply for the bioeconomy and climate change mitigation goals.

5        There are several models designed for calculation of soil temperatures (e.g., Yin and Arp, 1993; Rankinen et al., 2004; Jansson, 2012; Barrere et al., 2017; Park et al., 2017). In principle, the models approximate the solutions of differential equations describing water and heat flow. In conjunction with climate model data, these models can be utilized in evaluating the climate change impacts on soil temperature and frost conditions (e.g., Sinha and Cherkauer, 2010; Houle et al., 2012; Jungqvist et al., 2014; Oni et al., 2017). In addition to air temperature, the soil frost formation is affected by soil properties

like heat capacity and thermal conductivity. As well, snow as an efficient insulator of heat flow has large influence on soil frost. Snow depth in a spatially varying terrain varies even within short distances depending on vegetation and topography leading to variations in soil frost depth.

       In this study, we used a relatively simple soil temperature model developed originally by Rankinen et al. (2004). The only meteorological variables needed in the model calculations were daily mean air temperature and snow depth. Our objective

was to study the impacts of projected climate warming by 2100 on soil frost conditions, and consequently, on bearing capacity of different forest and soil types in Finland with regard to wintertime wood harvesting conditions and transport availability on forest truck roads. We used the soil temperature model in evaluating the soil frost conditions which largely define the ground-bearing capacity in winter. The ground-bearing capacity was assumed to be good if the depth of soil frost was at least 20 cm or depth of snow cover was at least 40 cm. First, we calibrated the model parameters, by describing, e.g. soil thermal

conductivity and specific heat capacity of soil for different soil types based on soil temperature observations from several stations across Finland. The effect of forest density on snow cover was also taken into account. Then, we evaluated the wintertime bearing capacity in future climatic conditions by using daily data from several global and regional climate model simulations downscaled onto an approximately 10 km × 10 km grid. The used global climate model (GCM) data were extracted from the Coupled Model Intercomparison Project phase 5 (CMIP5) database (Taylor et al., 2012) while the used regional

climate model (RCM) simulations were constructed within the EURO-CORDEX project (Jacob et al., 2014). The climate simulations were extended until 2099, considering two representative concentration pathway (RCP) scenarios, RCP4.5 and RCP8.5 (van Vuuren et al., 2011). We used data from wide set of climate models under the two emission scenarios to achieve a comprehensive picture of possible future outcomes. To foster the use of our results in forestry applications, the multi-GCM data describing the bearing capacity in different forest stands over different periods has been made publicly available.

## 2 Material and methods

### 2.1 Description of soil temperature model and its parametrization and validity

#### 2.1.1 Description of soil temperature model

Soil temperatures were calculated by using an extended version of soil temperature model originally introduced by Rankinen
et al. (2004). The model is derived from the law of conservation of energy and mass assuming constant water content in the
soil. This assumption simplified the model considerably with expense of its validity under extremely wet and dry conditions.
According to the model, soil temperature at depth $Z_S$ can be calculated as follows:

$$T_Z^{t+1} = T_Z^t + \frac{\Delta t \cdot K_T}{C_A \cdot (2 \cdot Z_S)^2} \cdot [T_{AIR}^t - T_Z^t], \tag{1}$$

where $T_Z^t$ (°C) is the soil temperature on a previous day, $T_{AIR}$ (°C) is the air temperature, $\Delta t$ is the length of a time step (s), $K_T$
(W m$^{-1}$ °C$^{-1}$) is the thermal conductivity of the soil and $C_A$ (J m$^{-3}$ °C$^{-1}$) is the heat capacity of the soil. $C_A$ can be approximated
as follows:

$$C_A \approx C_S + C_{ICE}, \tag{2}$$

where $C_S$ (J m$^{-3}$ °C$^{-1}$) is the specific heat capacity of the soil and $C_{ICE}$ (J m$^{-3}$ °C$^{-1}$) is the specific heat capacity due to freezing
and thawing. When $T_Z^t > 0$ °C, the latter term equals to 0.

As Eq. (1) did not take the insulating effect of snow cover into account, the equation was extended by an empirical relationship
(Rankinen et al., 2004):

$$T_Z^{t+1} = T_Z^t + \frac{\Delta t \cdot K_T}{(C_S + C_{ICE}) \cdot (2 \cdot Z_S)^2} \cdot [T_{AIR}^t - T_Z^t] \cdot [e^{-f_S \cdot D_S}], \tag{3}$$

where $f_S$ (m$^{-1}$) is an empirical damping parameter and $D_S$ (m) is snow depth. This model assumed that there is no heat flow
below the soil layer of consideration. To extend the model, Jungqvist et al. (2014) added parameters controlling the lower soil
thermal conductivity $K_{T,LOW}$ (W m$^{-1}$ °C$^{-1}$), lower soil specific heat capacity $C_{S,LOW}$ (J m$^{-3}$ °C$^{-1}$), and lower soil temperature $T_{LOW}$
(°C):

$$T_Z^{t+1} = T_Z^t + \frac{\Delta t \cdot K_T}{(C_S + C_{ICE}) \cdot (2 \cdot Z_S)^2} \cdot [T_{AIR}^t - T_Z^t] \cdot [e^{-f_S \cdot D_S}] + \frac{\Delta t \cdot K_{T,LOW}}{(C_{S,LOW} + C_{ICE}) \cdot 2 \cdot (Z_l - Z_S)^2} \cdot [T_{LOW} - T_Z^t], \tag{4}$$

where $Z_l$ (m) is the depth where $T_{LOW}$ prevails.

We assumed $T_{LOW}$ to be equal to the mean 2-m air temperature of previous 365 days and the values of parameters $K_T$, $C_S$, $C_{ICE}$,
$f_S$, $K_{T,LOW}$, $C_{S,LOW}$ and $Z_l$ were calibrated based on soil temperature observations (see the detailed description in the Section
2.1.2). According to forest harvesting specialists, 20 cm thick layer of frozen soil or 40 cm thick snow cover makes the terrain
passable for heavy harvesters even in soil types characterized by low bearing capacity (Eeronheimo, 1991). Keeping this in
mind, the emphasis in calibrating the parameters was given near the surface. Moreover, the parameters controlling heat flow
below the soil layer under consideration had only negligible effect on modelled soil temperatures near the surface.

### 2.1.2 Parametrization of soil temperature model

In the calibration of the parameters, we used soil temperature observations from the stations listed in Table 1. The idea was to search for typical values for the parameters in different soil types. The model was allowed to spin up for 1 year to reach thermal equilibrium in all of our calculations. Soil temperatures were measured every fifth day, except from Lettosuo (Korkiakoski et al., 2017), Apukka, Lompolojänkkä (Aurela et al., 2015) and Kaamanen (Aurela et al., 2001) the measurements were available on a daily basis. However, there were some time periods with missing data at these sites. The stations represented different soil types. The soil types were extracted from Soveri and Varjo (1977) and Heikinheimo and Fougstedt (1992), except for Lettosuo, Apukka, Lompolojänkkä and Kaamanen stations. According to the soil type map provided by the Geological Survey of Finland, the soil type at Apukka station is till. The Lettosuo station is situated in a drained peat bog and the stations Lompolojänkkä and Kaamanen are located on open fens (minerotrophic peatlands). Snow depth measurements needed in the calculations were not available from Lettosuo, Lompolojänkkä and Kaamanen stations and at these sites, snow depths measured on nearby stations were thus used in the model calibration. The replacement stations were Jokioinen for Lettosuo, Kenttärova for Lompolojänkkä and Inari for Kaamanen. Daily mean temperatures used in the model calibration were extracted from a gridded data set covering Finland (Aalto et al., 2016).

First, the parameter values were calibrated for each station and at different measurement depths using a Monte Carlo approach. The sampling ranges for the parameters (Table 2) were adopted from Jungqvist et al. (2014) but for $K_T$ the upper limit was extended from 1 W m$^{-1}$ K$^{-1}$ to 2 W m$^{-1}$ K$^{-1}$ to better represent the range of soil types and measurement depths considered in our study.

During the first calibration round, the soil temperature model was run 10 000 times for each station and measurement depth, sampling a new set of randomized parameters for each run from the chosen parameter ranges. Then, the set of parameters indicating the highest linear correlation between the observed and modelled soil temperatures at each station and each measurement depth was selected. Table S1 shows the calibrated values with their standard deviations after the first calibration round, as averaged over all the validation stations at 10 cm and 20 cm depths. At this point, $Z_l$ and $f_S$ without clear physical connection to local soil properties were set to their final values. Calibrated $Z_l$ values varied rather randomly within the sampling range implying that it was only marginally important parameter. The average for calibrated $Z_l$ values over all the stations and measurement depths was 6.8 m and $Z_l$ was set to that value. The calibrated values of $f_S$ varied between 9 and 10 with soil depths below 50 cm except at two stations. With increasing soil depth, calibrated $f_S$ values tended to generally decrease. Keeping in mind that we were most interested of the depths up to 20 cm, we set $f_S$ to 9.0.

During the second calibration round, the soil temperature model was run additional 100 000 times with the fixed $Z_l$ and $f_S$ values while the other parameters were sampled again. After this second calibration round, all other parameters except $K_T$ were also set to their final values. $K_{T,LOW}$ and $C_{S,LOW}$ were given the same values at all depths and locations as there were no clear relationship between their calibrated values and measurement depth or soil type, most likely because the heat flow from $Z_l$ was only marginally important compared to the heat flow from the surface, particularly near the surface. $C_{ICE}$ was set

to 11.0 J m$^{-3}$ K$^{-1}$ except at Sodankylä and Kevo with sandy soil to 8.0 J m$^{-3}$ K$^{-1}$. Calibrated $C_S$ values were mainly close to the lower limit of sampling range with depths below 100 cm while near the surface calibrated values were clearly higher. Thus, $C_S$ was set to depend on the soil depth following asymmetrical sigmoid function by using the calibrated values averaged across all the stations and measurement depths.

5       Then, the soil temperature model was run once more 10 000 times to sample only the $K_T$ values. In the final phase of the calibration, we sampled only the $K_T$ values as $K_T$ is clearly the most sensitive parameter in the soil temperature model (Jungqvist et al., 2014). The calibrated $K_T$ values tended to increase with soil depth at each location (Fig. 1). Anjala, Sodankylä and Lettosuo stations were selected to represent clay/silt, sand and peat soil types, respectively. At these stations, there were not much variability in soil type with different depths and calibrated $K_T$ values steadily increased with increasing soil depth.

Moreover, there were no missing observation depths at these stations. For the three soil types, $K_T$ was then estimated by fitting a logistic regression curve on the calibrated values on these three representative stations (Fig. 1). The final calibrated parameters used in the soil temperature calculations describing clay/silt, sand and peat soils are listed in Table 3.

      In addition, we modelled the soil frost in forest truck roads. In this case, we assumed that there is no snow on the surface and the parameters describing soil properties were set by giving 1/3 weight for the parameters used to describe sandy

soils and 2/3 for those describing clay/silt soils.

### 2.1.3 Validity of the modelled soil temperatures

Apart from the three stations (Lettosuo, Anjala and Sodankylä) used in calibration of $K_T$ in the final phase of model calibration , the modelled soil temperatures for clay/silt and sand soil types typically explained 90–99% of the observed variability in soil temperatures between the depths of 20 and 100 cm (Table S2). Near the surface the modelled temperatures correlated slightly

worse with the observed ones, as well as below 1 m. In spite of the generally high correlations, the modelled number of days with soil temperatures below 0 °C were still greatly overestimated, even by more than twofold on many stations (not shown). Thus, we also tested setting the model parameters by calibrating the modelled number of days with soil temperatures below 0 °C but then the correlations between observed and modelled soil temperatures became clearly worse, $R^2$ values dropping below 0.9 even at the best (not shown). In order to estimate more realistically the number of days with frozen soil, we thus assumed

in our model calculations that the soil does not freeze completely until the soil temperature drops below –0.1 °C in sand or below –0.5 °C in other soil types as some supercooling in the soil is needed to initiate the process of freezing (Kozlowski, 2009). For instance, in kaolinite clay ice lenses start to form in temperatures between –0.2 °C and –0.3 °C based on experiments and theoretical calculations (Style et al., 2011). At the depth of 20 cm, this reduced the number of soil frost days only by a few days in sand but roughly by one month in clay/silt and approximately by 1–3 months in peat. The choice of freezing points

was based on a study by Soveri and Varjo (1977) who stated that the freezing point in saturated sandy soil lies between 0 and –0.15 °C and in thin clay around –0.5 °C. Based on their study, in thick clay the freezing point can be as low as –20 °C, because the finer soil texture is, the stronger absorption and capillary water bound around the soil particles by reducing the freezing point. The melting point of soil was still set to 0 °C in all of our calculations.

## 2.2 Description of snow model and its parametrization and validity

### 2.2.1 Description of snow model

In order to estimate snow depth $D_S$ needed in the soil temperature calculations, we used a temperature index snow model based largely on approaches presented by Vehviläinen (1992). Meteorological variables needed in the snow depth calculations are
daily mean air temperature and daily total precipitation sum. First, the precipitation is divided into liquid and solid forms of precipitation as follows (Hankimo, 1976; Vehviläinen, 1992):

$P_{solid} = P_{tot}$, when $T_{mean} \leq -2.0\ °C$

$P_{solid} = \left(\frac{-T_{mean}}{8} + \frac{3}{4}\right) \cdot P_{tot}$, when $-2.0\ °C < T_{mean} \leq 0.0\ °C$

$P_{solid} = \left(\frac{-25T_{mean}}{90} + \frac{3}{4}\right) \cdot P_{tot}$, when $0.0\ °C < T_{mean} \leq 0.9\ °C$

$P_{solid} = \left(\frac{-5T_{mean}}{8} + \frac{17}{16}\right) \cdot P_{tot}$, when $0.9\ °C < T_{mean} \leq 1.3\ °C$            (5)

$P_{solid} = \left(\frac{-T_{mean}}{8} + \frac{33}{80}\right) \cdot P_{tot}$, when $1.3\ °C < T_{mean} \leq 3.3\ °C$

$P_{solid} = 0$, when $T_{mean} > 3.3\ °C$

$P_{liquid} = P_{tot} - P_{solid}$

where $P_{solid}$ (mm) is the amount of solid precipitation, $P_{liquid}$ (mm) is the amount of liquid precipitation, $P_{tot}$ (mm) is the total
amount of precipitation and $T_{mean}$ (°C) is the 2-metre daily mean air temperature.

      The used snow model calculates the snow water equivalent (SWE) and density of snowpack. SWE (mm) is divided into two components as follows:

$SWE = SWE_{new} + SWE_{old}$            (6)

where $SWE_{new}$ (mm) is the amount of SWE accumulated on the day considered and $SWE_{old}$ (mm) describes the amount of
snowpack left from the previous day. $SWE_{new}$ is calculated as follows:

$SWE_{new} = cps \cdot P_{solid} + SWE_{inc,liq}$            (7)

where cps is a correction factor for solid precipitation and $SWE_{inc,liq}$ (mm) is the increase of water storage in snowpack due to liquid precipitation. $SWE_{inc,liq}$ is limited by the water retention capacity of snowpack (WH) which is proportional to the total amount of snowpack and is thus determined as follows:

$WH = a \cdot SWE_{old}$            (8)

where $a$ is an empirical coefficient. $SWE_{inc,liq}$ is furthermore defined as follows:

$SWE_{inc,liq} = P_{liquid}$, when $P_{liquid} \leq WH$

$SWE_{inc,liq} = WH$, when $P_{liquid} > WH$            (9)

Decrease of SWE is caused both by evaporation from snowpack and by melting. Snowmelt is caused by thaw and liquid
precipitation. Rainfall affects snowmelt directly by heating snowpack but more importantly, also by creating drains in the snowpack and accelerating the ripening process of snow cover. $SWE_{old}$ is then calculated as follows:

$$\text{SWE}_{old}^{t+1} = \text{SWE}_{old}^t + \text{SWE}_{new}^t - [\text{km}_t \cdot (T_{mean}^t - \text{tm}) - \text{pm} \cdot P_{liquid}^t \cdot (T_{mean}^t - \text{tm}) - \text{ev}] \cdot \Delta t \qquad (10)$$

where km (mm $°C^{-1}$ $d^{-1}$) is a degree-day factor, tm (°C) is threshold air temperature for snowmelt, pm ($°C^{-1}$ $d^{-1}$) is a melt factor related to liquid precipitation and ev (mm $d^{-1}$) is evaporation from snowpack. The degree-day factor km is calculated as follows (Anderson, 1973):

$$\text{km} = \frac{\text{kmax} + \text{kmin}}{2} + \sin\left(\frac{2N \cdot \Pi}{366}\right) \cdot (\text{kmax} - \text{kmin}) \qquad (11)$$

where kmax (mm $°C^{-1}$ $d^{-1}$) is the degree-day factor on June 21st, kmin (mm $°C^{-1}$ $d^{-1}$) is the degree-day factor on December 21st and $N$ is the day number beginning with March 21st.

Density of snow is calculated separately for new and old snow. Density of freshly fallen snow ($\rho_{s,new}$) is calculated as follows:

$$\rho_{s,new} = b \cdot T_{mean} + c, \text{ when } \rho_{s,new} \geq \rho_{s,new_{min}} \qquad (12)$$

where $b$ (kg $m^{-3}$ $°C^{-1}$) and $c$ (kg $m^{-3}$) are empirical coefficients and $\rho_{s,new_{min}}$ (kg $m^{-3}$) is the minimum possible density of freshly fallen snow.

Density of old snow ($\rho_{s,old}$) is increased due to aging, thawing and liquid precipitation and is thus calculated as follows:

$$\rho_{s,old}^t = \rho_s^{t-1} + (\rho_{s,inc} + \rho_{s,inc,rain} \cdot P_{liquid}) \cdot \Delta t, \text{ when } \rho_{s,old}^t \leq \rho_{s_{max}} \qquad (13)$$

where $\rho_s^{t-1}$ (kg $m^{-3}$) is the density of snowpack on a previous day, $\rho_{s,inc}$ (kg $m^{-3}$ $d^{-1}$) is the density increment due to aging and thawing of snowpack, $\rho_{s,inc,rain}$ (kg $m^{-3}$ $mm^{-1}$ $d^{-1}$) is the density increment due to liquid precipitation and $\rho_{s_{max}}$ (kg $m^{-3}$) is the maximum possible density of snowpack. $\rho_{s,inc}$ (kg $m^{-3}$ $d^{-1}$) is defined as follows:

$$\rho_{s,inc} = \rho_{s,inc,age}, \text{ when } T_{mean} \leq 0 \text{ °C}$$

$$\rho_{s,inc} = \rho_{s,inc,age} + \rho_{s,inc,thaw} \cdot T_{mean}, \text{ when } T_{mean} > 0 \text{ °C} \qquad (14)$$

where $\rho_{s,inc,age}$ (kg $m^{-3}$ $d^{-1}$) is a coefficient defining the density increment of snowpack due to aging and $\rho_{s,inc,thaw}$ (kg $m^{-3}$ $°C^{-1}$ $d^{-1}$) is a coefficient related to the density increment of snowpack due to thawing.

Finally, $D_S$ (m) is calculated as follows:

$$D_S = \frac{\text{SWE}_{new}}{\rho_{S,new}} + \frac{\text{SWE}_{old}}{\rho_{S,old}} \qquad (15)$$

### 2.2.2 Parametrization of snow model

Parameters for the snow model were calibrated with a similar manner as for the soil temperature model. We randomly sampled 10 000 times the parameters $a$, $b$, $c$, cps, tm, pm, ev, kmax, kmin, $\rho_{s,new_{min}}$, $\rho_{s_{max}}$, $\rho_{s,inc,rain}$, $\rho_{s,inc,age}$ and $\rho_{s,inc,thaw}$ from the parameter ranges shown in Table 4. Then, the model was ran with each of these 10 000 set of parameters for the seven stations with soil temperature observations covering the period 2007–2014 (Table 1). The snow model was run over the period 1961–2014 by using the Finnish gridded climate data (Aalto et al., 2016) and the period 2006–2014 was used as the calibration period for the snow model. We minimized the root-mean-square error (RMSE) between modelled and observed snow depths on the stations during the calibration period by selecting the set of parameters indicating the smallest RMSE on each station. Then,

the calibrated parameters were averaged among all the seven stations to give the final parameters for the snow model (Table 4). Exceptions were kmax and kmin which seemed to show a latitudinal dependence as expected. These parameters were thus approximated by latitudinal-dependent exponent functions.

During the calibration period 2006–2014, the snow model with calibrated parameters shown in Table 4 explained 94–96% of the observed variability in snow depth except at Apukka, where $R^2$-value was only 0.84 (Table S3). When using the parameters calibrated for each station before averaging, the $R^2$-values were on average approximately 0.01 higher (not shown). We also tested the model with fixed kmax and kmin values averaged similarly as the other parameters and then the $R^2$-values were on average 0.003 lower than those showed in Table S3. Except at Apukka, the model performance was in this case slightly worsen at every station.

As the snow depth measurement sites are located on open environments, the calibrated parameters shown in Table 5 were used to model snow depth on open habitats. In forested areas, snow cover is reduced due to interception by the canopy, evaporation of the intercepted snow and enhanced wintertime snowmelt below the canopy (Hedstrom and Pomeroy, 1998; Varhola et al., 2010). Interception typically increases with increasing forest density and leaf area index (Lundberg and Koivusalo, 2003; Rasmus et al., 2013). Interception can be as high as nearly 50% of precipitation (Stähli and Gustafsson, 2006). In order to model the soil frost in different kind of forest stands, we added an interception coefficient to the snow model. In addition to open habitats, the calculations were performed for forests with three different density classes corresponding roughly to deciduous forest or sparse mixed forest, pine forest and dense spruce forest. The interception coefficients for these forest stands were extracted from Lundberg and Koivusalo (2003). To reduce the modelled snow cover in forests, SWE$_{new}$ was multiplied with the interception coefficient in every time step.

Forest canopy also shelters snow cover from direct sunlight which reduces the degree-day factor. In general, the melting proceeds more slowly the denser is the forest. Vehviläinen (1992) presented experimental degree-day factors for open and forested areas for different river catchments and also based on earlier studies for both open areas and for different kind of forests (Gurevich, 1950; Hiitiö, 1982). Based on Vehviläinen (1992), the degree-day factor is typically 30–60% smaller in forests compared to open areas, depending on river catchment and the time of melting season but the estimates were quite variable. Hiitiö (1982) concluded that the degree-day factor in spruce forests is reduced by 28% from its value in open areas whereas Gurevich (1950) suggested over 60% reduction in dense spruce forests. In general, the reduction is smaller in the beginning of the melting season as solar radiation increases towards summer. We used rough estimates for the degree-day factor in different forest types based on the literature review presented by Vehviläinen (1992).The coefficients used in reducing kmax and kmin as well as the interception coefficients used in this study for different forest types are shown in Table 5.

**2.2.3 Validity of the modelled snow depths**

The validation period 1962–2005 was divided into two sub-periods, 1962–1980 and 1981–2005, because the precipitation gauges and their wind shields in Finland were changed during 1981–1982 in order to improve catch efficiency of snow. Before 1981, Nipher-shielded Wild gauges were used and after 1982 shielded Tretyakov gauges which are known to suffer less from

wind-induced undercatch of snowfall (Yang et al., 1999; Taskinen and Söderholm, 2016). During the period 1981–2005, the $R^2$-values varied between 0.87 and 0.93, so the model performance was somewhat lower than during the calibration period. Before 1981, the model performance was even worse except at Apukka where the snow model performed best during the validation periods. This is probably related to the fact that the modelled snow depths in a grid cell surrounding the Apukka

station were on average 31% higher than the observed ones at the station during the calibration period while at other stations the modelled and observed snow depths resembled each other more closely (Table S3). The overestimation of snow depth at Apukka might be related to local microclimatological characteristics as we used the gridded climate data in snow modelling, not the station observations. Already at Rovaniemi Airport weather station, located only at a distance of 7.9 km from the Apukka station, the observed snow depths tend to be remarkable higher (not shown). It is moreover noteworthy that modelled

snow depths were in general systematically underestimated during the validation periods, especially before 1981. This indicates that a higher correction factor cps should have been used for the previous decades to improve the model performance due to the higher undercatch of snowfall before 1981 (Taskinen and Söderholm, 2016).

## 2.3 Simulation of soil frost and snow depth for different forest and soil types under changing climatic conditions

The soil frost and snow depth calculations were performed for each possible combinations of three soil types and four forest

types to evaluate the changes in soil bearing capacity. In addition, the calculations were performed for forest truck roads without snow cover leading to a total of 13 different combinations of soil and forest types. Calculations for each of these soil and forest types were performed on every grid cell. The simulation results were analysed for the near-future period 2021–2050 and for the far-future period 2070–2099 as compared to the baseline period 1981–2010. In addition, over the baseline period the soil temperature and snow depth models were ran also by using the observational Finnish gridded climate data (Aalto et

al., 2016). We modelled the number of days with good bearing capacity in the forest harvesting point of view. Good bearing capacity was assumed to prevail when the soil frost penetrated continuously from the surface to at least the depth of 20 cm or when the snow depth exceeded 40 cm.

The calculations for the period 1980–2099 under changing climate were completed using daily data from six GCMs (listed in Table S4) participating in the CMIP5 (Taylor et al., 2012; Flato et al., 2013). In addition, we used daily data from 11

bias-adjusted RCM simulations (listed in Table S5) constructed within the EURO-CORDEX project (Jacob et al., 2014). The variables used in this study were daily mean 2-m air temperature and daily precipitation sum.

The GCMs were chosen on the basis of their skill to simulate present-day average monthly temperature and precipitation climatology in northern Europe. However, the GCM outputs are always more or less biased and presented on a relatively coarse grid. Hence, before soil temperature calculations, we performed for the GCM data a combined bias correction

and statistical downscaling from a lower to a higher resolution (Maraun and Widmann, 2018). In this procedure, the distributions of downscaled weather variables were modified to correspond the observed distributions within the calibration period (1981–2010 in our case) in the resolution of the observational data set. Then, the same corrections were applied to the whole simulation period. As our observational data set, we used the Finnish gridded climate data on a regular 0.1°×0.2° grid

(Aalto et al., 2016). The combined statistical downscaling and bias correction was performed by applying a quantile mapping technique using smoothing. A detailed evaluation of this procedure for correcting simulated temperature time series was presented by Räisänen and Räty (2013) and for correcting simulated precipitation time series by Räty et al. (2014).

From the EURO-CORDEX archive we chose the set of models with the largest number of simulations available with a uniform bias-adjustment approach. All the used RCM simulations were constructed in the Institut Pierre Simon Laplace (IPSL) using a cumulative distribution function (CDF) method (Vrac et al., 2016). They were downscaled onto the EUR-11 CORDEX domain having a horizontal resolution of ~0.11°×0.11° by using the Water and Global Change Forcing Data ERA Interim (WFDEI) meteorological forcing data set (Weedon et al., 2014) over a calibration period 1979–2014. Before soil temperature calculations, we linearly interpolated the RCM data onto the same 0.1°×0.2° grid with the GCM data.

Both GCM and RCM model ensembles were based on the RCP4.5 and RCP8.5 scenarios (van Vuuren et al., 2011), which are widely used in climate change impact studies. The RCP4.5 scenario represents a world characterized by relatively well succeeded mitigation of greenhouse gas emissions and in that scenario, the radiative forcing stabilizes at 4.5 W m$^{-2}$ in 2100. The RCP8.5 scenario, on the other hand, represents a world without any efficient mitigation activities applied and leads to almost twice as large radiative forcing and climate warming on the global scale by 2100. In Finland, the projected increases in mean annual temperature and precipitation are under the RCP8.5 scenario up to 6 °C and 18%, respectively, when the atmospheric $CO_2$ concentration approaches 1000 ppm by 2100 (Ruosteenoja et al., 2016). The increases in temperature and precipitation are both predicted to be clearly higher in winter months than in summer.

## 3 Results

### 3.1 Wintertime bearing capacity during the baseline period 1981–2010

The modelled annual average number of days with good wintertime bearing capacity during 1981–2010 based on observational weather data in three different forest types common in Finland, i.e. dense Norway spruce (*Picea abies*) forests on clay or silt soil, Scots pine (*Pinus sylvestris*) forests on sandy soil and Scots pine forests on peatlands is showed in the left panel of Fig. 2. The number of days with good wintertime bearing capacity on forest truck roads is shown as well. Upland forests on sandy soil generally have most and forests on drained peatlands least days with good bearing capacity as the soil frost penetrates fastest in sand and slowest in peat. The winter period with good bearing capacity lasts in northern Finland on average approximately 5–7 months, depending on forest and soil type. In the central parts of the country, the wintertime bearing season lasts about 3–4 months on drained peatlands and roughly about 5 months on other soil types. In the coastal areas of southern and southwestern Finland, the length of bearing season varies typically between 2 and 4 months per winter depending on the soil type. On forest truck roads, the bearing season is modelled to last about 3–4 months per winter in southern Finland and about half a year in northernmost Finland.

The used models generally reproduce the spatial pattern of wintertime bearing season length during the baseline period as expected as the model data has been bias-corrected. In the GCM ensemble, the difference in the number of days with

good bearing capacity between the multi-model ensemble and model calculations using observational weather data tend to be almost everywhere even less than 5 days, except in pine-dominated drained peatland forests where the bearing season length is locally overestimated by 20 days (Fig. 2h). In the RCM ensemble, the agreement between the calculations using model data and observational weather data is generally poorer but the difference in the number of days with good bearing capacity is still less than 10 days over most of Finland.

## 3.2 Wintertime bearing capacity during the future periods 2021–2050 and 2070–2099

The projected change in the average wintertime bearing season length for the above-mentioned forest types and for forest truck roads is displayed in Fig. 3 on the basis of the GCM ensemble and in Fig. 4 based on the RCM ensemble. The wintertime bearing season is projected to shorten roughly by about one month for the near-future period 2021–2050. The change is only a little smaller under the RCP4.5 than RCP8.5 scenario. The GCM and RCM ensembles indicate rather similar changes. The projected change is moreover rather similar among the different soil types.

For the far-future period 2070–2099 the projected shortening of the wintertime bearing season is clearly more pronounced. In addition, the difference in the magnitude of change between the two forcing scenarios becomes larger. If the high-emission RCP8.5 scenario will be realized, the bearing season may shorten by more than 3 months over large parts of the country. On drained peatlands, the change remains smaller in southern Finland as the bearing season lasts there less than 3 months per winter already during the baseline period. On the other hand, in upland forests on sandy and clay or silt soil types the projected shortening of bearing season is largest in southern and western Finland. In these areas, the bearing season is projected to shorten by about 2 months also under the RCP4.5 scenarios.

The projected change in the wintertime bearing season length on forest truck roads accompanies the projected change on different forest types, especially on sandy and clay or silt soil types. By mid-century, the wintertime bearing season on forest truck roads may shorten by more than one month in western Finland. In the end of the 21st century, the bearing season on forest truck roads may last even on average only about one month per winter in the southwestern parts of the country of the RCP8.5 scenario will be realized.

## 3.3 Interannual variability in the wintertime bearing season length

Interannual variability in the wintertime bearing season length is illustrated in Fig. 5. Here we show the results only derived from the GCM ensemble and only for pine forests on drained peatlands as they are the most difficult sites for forest harvesting. During the baseline period, the bearing season length exceeds one month on more than 80% of the winters except in the coastal areas in southern and southwestern Finland (Fig. 5d). In Lapland, the bearing season lasts even on the mildest winters 2–3 months but at the southwestern coast, the mildest winters do not express good bearing capacity on any day (Fig. 5c). During the near-future period 2021–2050, the share of winters with less than one month of good bearing capacity is projected to somewhat increase, particularly in southern and western Finland (Figs 5h and 5l). However, over most of Finland a large majority of winters still have a sufficient amount of days with good bearing capacity, although the conditions during the mildest

winters are projected to become more difficult. For the far-future period 2070–2099, the situation is projected to change more considerably, particularly if the RPC8.5 scenario will be realized (Figs 5q-5t). Based on the multi-GCM mean, only a few winters express longer than one month bearing season in southern and western Finland (Fig. 5t). Even in eastern Finland and southwestern Lapland the bearing season length is projected to exceed one month approximately only on every other winter.

During the mildest winters, soil frost may penetrate to 20 cm or snow depth exceed 40 cm only on localized areas in northern Finland.

### 3.4 Intermodel variability in the projected wintertime bearing season length

In Figs 6 and 7 we illustrate the range of possible outcomes between different climate model projections for wintertime bearing season length in pine forests on drained peatlands. Fig. 6 shows the highest and lowest annual mean number of days with good

wintertime bearing capacity among the used climate models on each grid cell over the two 30-year future periods under the two forcing scenarios, separately for the 6 GCMs and for the 11 RCMs. Fig. 7 displays for the both model ensembles the annual mean number of days with good wintertime bearing capacity with multi-model standard deviations as averaged over whole of Finland. It is visible that already during the near-future period 2021–2050, the projected average wintertime bearing season length diverges by more than one month in both model ensembles. During the far-future period 2070–2099, the range

in the average bearing season length among the model projections is typically already more than two months. The models with strongest warming even indicate that the average bearing season length in the end of the 21st century might be 0 days at the southwestern coast of Finland meaning that even during the coldest winters soil frost would not penetrate to 20 cm. The coldest model projections, on the other hand, indicate that the wintertime bearing season length would shorten only by about one month by the end of the 21st century. Also Fig. 7 confirms that the projected changes are rather similar in both model ensembles

and during the near-future period also under both forcing scenarios.3.5 Relative importance of soil frost and snow cover in providing good wintertime bearing capacity

As the bearing season length is affected both by soil frost and snow cover, it is worth of inspecting projected changes in these two variables separately. During a typical winter, snow depth exceeds the limit of 40 cm in eastern and northern Finland for several months but in the coastal areas in southern and western parts of the country, snow depth rarely exceeds 40

cm. Thus, in western Finland the bearing season length is mainly controlled by soil frost (Fig. 8). In the east, on the contrary, good bearing capacity is more often provided only due to the thick snow cover. In northern Finland, despite of the thick snow cover, also soil frost penetrates in most areas typically deep enough to assure good bearing capacity. The spatial picture is projected to remain similar during the present century but the cases with deep snow cover with less than 20 cm of soil frost seem to become slightly less abundant and almost non-existent in western Finland.

## 4 Discussion and conclusions

### 4.1 Evaluation of methodology

The bearing capacity of forest soils was evaluated on the basis of soil temperature model that had been previously applied successfully in Finnish and Swedish conditions (Rankinen et al., 2004; Jungqvist et al., 2014). In this study, the model parameters were calibrated separately for three different soil types based on soil temperature observations. Typically the model explained 90–99% of the observed soil temperature variability. However, on most of the locations the model tended to overestimate the frost formation and soil temperature variations near the surface. Thus, the relative importance of snow cover in providing good wintertime bearing capacity is assumedly larger than showed in Fig. 8.

Several assumptions were made in this study in order to simplify the calculations. Firstly, 20 cm depth of soil frost or 40 cm depth of snow cover may not be sufficient for good bearing capacity in all soil conditions. This is because the required soil frost depth is dependent on soil wetness, for example. In this study, we assumed constant water content in the soil. In dry soil conditions more than 50 cm of soil frost may be required to carry 10-ton trucks (Shoop, 1995). However, in the present study main focus was on carrying capacity of drained peatlands which are the wettest forest environments in Finland and thus most difficult to harvest wood in summer. Besides the experiment based estimate of Eeronheimo (1991), about 20 cm depth of soil frost has been found sufficient to ensure the bearing capacity of soil for forest harvesters in Finnish conditions also in model based studies (Suvinen et al., 2006; Kokkila, 2013).

The effect of forest density on soil frost formation was taken into account in our study in the modelling of snow depth. The snow model was first calibrated for open areas similarly as the soil temperature model. The effect of forest density on the snow depth was then evaluated based on literature. As we did not have any snow depth measurements from forested sites, the modelled snow depths for forested areas were susceptible for biases. In general, snow depth decreases with increasing forest density. In our calculations this led to somewhat enhanced soil frost formation in denser forests. However, the differences between different forest types were small. In reality, forest vegetation also acts as an insulator. Thus, open areas often have deeper soil frost than forests despite of having also deeper snowpack, but the results from different sites are contradictory (Soveri and Varjo, 1977). According to Yli-Vakkuri (1960) soil frost penetrates particularly deep in dense spruce forests due to their large canopy cover leading to shallow snow depths.

The climate change impact on wintertime bearing capacity of forest soils was taken into account by using climate model simulations. The climate models usually poorly simulate soil frost penetration (e.g., Sinha and Cherkauer, 2010). Moreover, in northern Europe most of the GCMs and RCMs have a cold bias in winter (Cattiaux et al., 2013; Kotlarski et al., 2014). Thus we did not apply soil temperature or snow depth outputs directly from the models, but first bias-corrected and downscaled the climate model data onto a $0.1° \times 0.2°$ (approximately 10 km $\times$ 10 km) grid. Then, we used a relatively simple land surface model that could be calibrated for different soil types and implemented with the available data to calculate the soil temperatures. By combining the soil and vegetation information with the soil frost calculations, the expected changes in timber harvesting conditions can be evaluated in a relatively small scale. However, in reality there is considerable variability

in the soil frost conditions also within relatively similar soil and vegetation types. For example, the level of groundwater has a substantial impact on the soil frost depth (Soveri and Varjo, 1977). These kind of small-scale variations could not be taken into account in our approach although the results are presented in a relatively high-resolution grid.

In all, there are several sources of uncertainty in the results of this study. The calibrated parameters describing different soil types are not exact and in reality they are never exactly equal in different locations. Moreover, a model with almost equally good fit could be constructed with very different set of parameters if the parameter values would be adjusted conveniently. This is because there is no single best model parameter set but many model state descriptions can generate equally good calibration outputs (Beven, 2006; Jungqvist et al., 2014). However, on many locations the model performed reasonably well even with the wrong soil type (Table S2) and as the stations used in calibrating the model are located in different parts of Finland, we assume that possible future changes in soil characteristics, including thermal conductivity, do not crucially change the results.

Despite of many uncertainties in soil frost modelling mentioned above, our results were in general reasonable. For instance, based on station observations of soil frost and snow cover depths Eeronheimo (1991) stated that the length of transporting season in peatland forests varies in Finland approximately between 60 and 190 days as defined based on the requirement of 20 cm of soil frost or 40 cm of snow cover on unfrozen ground. When comparing this to our results for bearing season length during the baseline period in drained pine-dominated peatland forests (Fig. 2g), it can be seen that the difference is mainly less than 15 days.

Considering the future projections, the two used climate model ensembles we used yielded rather similar results (Figs. 3 and 4) including increasing scatter among the model projections towards the end of the century (Fig. 6). The RCM ensemble using WFDEI forcing data set (Weedon et al., 2014) in bias correction had some differences in spatial small-scale features of bearing season length pattern compared to the GCM ensemble that used in bias correction the gridded Finnish climate data set (Aalto et al., 2016). Most notably, the RCMs produced longer soil frost periods along the coast of Bothnian Bay. Nevertheless, both model ensembles reproduced after the bias correction satisfactorily the general large-scale pattern of bearing season length when compared to the results calculated from observation-based gridded climate data (Fig. 2).

**4.2 Evaluations of main results and their implications to forestry**

In accordance with previous studies (Venäläinen et al., 2001a, 2001b; Kellomäki et al., 2010), our results suggest that climate warming will lead to shorter soil frost periods reducing wintertime ground-bearing capacity. Also a reduction in snow cover contributes to decreasing bearing capacity (Räisänen and Eklund, 2012). The projected decrease in the wintertime bearing season length was similar in the studied two climate model ensembles. Most likely the bearing season length in winter will decrease by about one month until mid-21st century and by about 1.5–3 months until 2100. Nevertheless, there is considerable variation in the rate of the projected change among the individual climate model simulations.

In relative terms, the decrease in the wintertime bearing season length is most prominent in southern and western Finland. That is because in Lapland the season is typically three months longer than at the southern coast and thus even the most extreme projections do not lead to a complete loss of the ground frost there. Similarly, abilities for wintertime logging on drained peatlands are expected to worsen more than on upland soil types. Based on our results it is evident that in the latter half of the century on drained peatlands logging cannot be expected to be conducted during frozen soil conditions in most parts of Finland. On the other hand, shortening of the soil frost season leads to an earlier transition to summer conditions. This leads to a reduced soil moisture content in spring and also in summer the soil moisture content is projected most likely to decrease (Ruosteenoja et al., 2018). Consequently, possibilities for summertime logging may improve.

Our results considering the climate change impact on the conditions of forest harvesting and logistics provide urgently needed support for the planning of wood harvesting and transportation in different timespans and regions. During the last couple of decades, there has also been a trend towards heavier machinery in forest harvesting (Ala-Ilomäki et al., 2011) and the allowed maximum weight of timber trucks has increased (e.g., Malinen et al., 2014). The forest truck roads in Finland have been mainly constructed between 1960 and 1990 and many of them need a major renovation before timber haulage can take place (Kaakkurivaara et al., 2005). Hence, maintaining sufficient bearing capacity on forest truck road network is also important. Fortunately, there are several possibilities to improve mobility of forest machinery on poorly bearing conditions. For example, the carrying surface can be extending by using auxiliary wheels, width of individual wheels can be widened, tyre pressure can be reduced or wider tracks can be used (Airavaara et al., 2008). One possibility is also to use two-stage wood harvesting. In this method, the cutting is conducted when the soil is still unfrozen but wood stacks are extracted later in winter when the soil is frozen (Heikkilä, 2007). Logging residues can be placed on the forwarding trails to improve the soil bearing capacity as is done in Northern Scotland on peatland harvesting (Röser et al. 2011). This, however, reduces the volume of harvestable logging residues for energy use. Anyway, as there is a pressure to increase wood harvesting in drained peatlands in the future with simultaneous decrease in the ground-bearing capacity, there is a clear need to develop new cost-effective solutions for peatland harvesting, taking into account this anticipated decrease in ground-bearing capacity.

### 4.3 Conclusions

The results of this study indicate clearly that the soil frost period in Finland will become shorter as climate becomes warmer. Hence, it is evident that a larger share of logging need to be carried out under unfrozen soil conditions. Particularly this holds for drained peatlands as the soil frost period is there shortest due to the insulating effect of peat. In southern and western Finland, drained peatlands might remain virtually frost-free on most of winters during the latter half of the current century. Already by 2050, the winters with only short frost periods will become more common. The projected decrease in the bearing capacity, particularly in drained peatlands, with simultaneously increasing demand for the wood utilization from peatlands induces a clear need for the development of new sustainable and efficient logging practices. To foster the use of our results,

the data showing the average bearing season length in different combinations of soil and forest types during different study periods will be made publicly available.

The results presented here will also serve as a basis for several future analysis. The effects of changing climate on timber procurement in different regions of Finland should be analysed in more detail. In addition, it should be analysed a what kind of supply technology development needs exist concerning logging machinery, transportation fleet and information systems managing and monitoring the raw material flows in the future. These soil frost calculations can be also applied in studying climate change impacts on wind damage risks to forests as soil frost makes trees more resistant for uprooting by anchoring them effectively to the ground (Peltola et al., 1999; Saad et al., 2017). With regard to harvesting logistics, it would be interesting to study also whether clear cutting facilitates the transformation of some peatland stands marked for cutting in winter into stands marked for cutting in summer (Ala-Ilomäki et al. 2011, Sirén et al. 2013, Laitila et al. 2016). This is because, compared to thinning, clear cutting allows greater freedom in the location of forwarding routes on site, as well as in organising route schedules (Uusitalo et al. 2015b).

## 5. Data availability

The climate model data used in this study can be downloaded from the CMIP5 and CORDEX archives, e.g. at https://esgf-node.ipsl.upmc.fr/projects/esgf-ipsl/. The observational gridded Finnish climate data from 1961 onwards can be downloaded from the Paituli spatial data download service at http://avaa.tdata.fi/web/paituli/metadata. The soil temperature observations from the stations listed in Table 1 are available on request from the corresponding author. The spatial data describing the multi-GCM mean wintertime bearing season length on different combinations of soil and forest types over the studied periods can be downloaded from the Paituli spatial data download service at https://avaa.tdata.fi/web/paituli/latauspalvelu?data_id=il_soil_conditions_1981_txt_wgs.

## Acknowledgments

This research has been supported by the Strategic Research Council at the Academy of Finland through the FORBIO (Sustainable, climate-neutral, and resource-efficient forest-based bioeconomy) research project (grant numbers 293380 and 314224). We acknowledge the World Climate Research Programme's Working Group on Regional Climate and Working Group on Coupled Modelling, former coordinating body of the CORDEX project and latter being responsible for CMIP. We are moreover grateful to all the modelling groups (listed in Tables S3 and S4 in the supplementary material of this paper) for producing and making their model outputs available. The model data used in this work were obtained from the Earth System Grid Federation portal. For CMIP the US Department of Energy's Program for Climate Model Diagnosis and Intercomparison provides coordinating support and led development of software infrastructure in partnership with the Global Organization for Earth System Science Portals. Kimmo Ruosteenoja is acknowledged for downloading and preprocessing the GCM data. Olle

Räty and Jouni Räisänen from Department of Physics, University of Helsinki, are acknowledged for developing the bias correction software applied on the GCM data. Annalea Lohila is acknowledged for providing soil temperature observations for Lettosuo, Kaamanen and Lompolojänkkä stations.

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

**Table 1.** Soil temperature measurement data used in the calibration of soil temperature model.

| Station name | Latitude | Longitude | Soil type | Soil temperature measurement depths | Observation period |
|---|---|---|---|---|---|
| Lettosuo | 60°38'31"N | 23°57'35"E | peat | 5, 15, 30 and 40 cm | 2009–2014 |
| Anjala | 60°41'47"N | 26°48'40"E | clay/silt | 10, 20, 30, 50, 70, 100, 150 and 200 cm | 2007–2014 |
| Jyväskylä | 62°23'51"N | 25°40'15"E | silt | 10, 20, 30, 70, 100, 150 and 200 cm | 2007–2014 |
| Ylistaro | 62°56'17"N | 22°29'20"E | silt/clay | 20, 50, 100, 200 and 300 cm | 2007–2014 |
| Maaninka | 63°8'36"N | 27°18'47"E | fine sand/silt | 10, 20, 30, 50, 70, 100, 150 and 200 cm | 2007–2014 |
| Apukka | 66°34'46"N | 26°0'40"E | till | 10, 20, 30, 50, 70, 100, 150 and 200 cm | 2007–2014 |
| Sodankylä | 67°21'60"N | 26°37'44"E | sand/gravel | 10, 20, 30, 50, 70, 100, 150 and 200 cm | 2007–2014 |
| Lompolojänkkä | 67°59'50"N | 24°12'33"E | peat | 5, 15 and 30 cm | 2007–2009 |
| Kaamanen | 69°8'26"N | 27°16'11"E | peat | 5, 15 and 30 cm | 2004–2012 |
| Kevo | 69°45'23"N | 27°0'24"E | sand | 10, 20, 50, 100 and 200 cm | 2007–2014 |

**Table 2.** Parameter ranges for the model calibration simulations.

| Parameter | Unit | Sampling range |
|---|---|---|
| Soil thermal conductivity, $K_T$ | W m$^{-1}$ K$^{-1}$ | 0...2 |
| Specific heat capacity of soil, $C_S$ | J m$^{-3}$ K$^{-1}$ | 0.5...3.5 ($10^6$) |
| Specific heat capacity due to freezing and thawing, $C_{ICE}$ | J m$^{-3}$ K$^{-1}$ | 4...15 ($10^6$) |
| Empirical snow parameter, $f_S$ | m$^{-1}$ | 0...10 |
| Lower soil thermal conductivity, $K_{T,LOW}$ | W m$^{-1}$ K$^{-1}$ | 0...1 |
| Lower soil specific heat capacity, $C_{S,LOW}$ | J m$^{-3}$ K$^{-1}$ | 0.5...3.5 ($10^6$) |
| Lower soil temperature depth, $Z_l$ | m | 3...15 |

**Table 3.** Calibrated parameters of soil temperature model for different soil types. The values of parameters are as follows: $a$=25.788, $b$=0.18029, $c$=0.71240, $d$=-0.58616, $e$=2.7183, $f$=0.33272, $g$=18.231, $h$=0.17401, $i$=1.0885, $j$=-1.0703, $k$=0.57526, $l$=19.217, $m$=0.18222, $n$=0.20835, $o$=1.8970, $p$=124.76, $q$=24.398, $r$=0.46356, $s$=0.010773, Z=soil depth (cm).

| Parameter | Clay/Silt | Sand | Peat |
|---|---|---|---|
| Soil thermal conductivity, $K_T$ (W m$^{-1}$ K$^{-1}$) | $K_T = c/(1 + ae^{-bZ})$, when Z < 8 cm; $K_T = d + f \ln Z$, when Z ≥ 8 cm | $K_T = i/(1 + ge^{-hZ})$, when Z < 11 cm; $K_T = j + k \ln Z$, when Z ≥ 11 cm | $K_T = n/(1 + le^{-mZ})$ |
| Specific heat capacity of soil, $C_S$ (J m$^{-3}$ K$^{-1}$) | $C_S = \left(r + \dfrac{o - r}{(1 + (Z/q)^p)^s}\right) \cdot 10^6$ | $C_S = \left(r + \dfrac{o - r}{(1 + (Z/q)^p)^s}\right) \cdot 10^6$ | $C_S = \left(r + \dfrac{o - r}{(1 + (Z/q)^p)^s}\right) \cdot 10^6$ |
| Specific heat capacity due to freezing and thawing, $C_{ICE}$ (J m$^{-3}$ K$^{-1}$) | $C_{ICE} = 11.0 \cdot 10^6$ Jm$^{-3}$K$^{-1}$ | $C_{ICE} = 7.0 \cdot 10^6$ Jm$^{-3}$K$^{-1}$ | $C_{ICE} = 11.0 \cdot 10^6$ Jm$^{-3}$K$^{-1}$ |
| Empirical snow parameter, $f_S$ (m$^{-1}$) | $f_S = 9.0$ m$^{-1}$ | $f_S = 9.0$ m$^{-1}$ | $f_S = 9.0$ m$^{-1}$ |
| Lower soil thermal conductivity, $K_{T,LOW}$ (W m$^{-1}$ K$^{-1}$) | $K_{T,LOW} = 0.8$ Wm$^{-1}$K$^{-1}$ | $K_{T,LOW} = 0.8$ Wm$^{-1}$K$^{-1}$ | $K_{T,LOW} = 0.8$ Wm$^{-1}$K$^{-1}$ |
| Lower soil specific heat capacity, $C_{S,LOW}$ (J m$^{-3}$ K$^{-1}$) | $C_{S,LOW} = 1.8 \cdot 10^6$ Jm$^{-3}$K$^{-1}$ | $C_{S,LOW} = 1.8 \cdot 10^6$ Jm$^{-3}$K$^{-1}$ | $C_{S,LOW} = 1.8 \cdot 10^6$ Jm$^{-3}$K$^{-1}$ |
| Lower soil temperature depth, $Z_l$ (m) | $Z_l = 6.8$ m | $Z_l = 6.8$ m | $Z_l = 6.8$ m |

**Table 4.** Parameter ranges for snow model calibration and the calibrated parameter values.

| Parameter | Unit | Sampling range | Calibrated value[1] |
|---|---|---|---|
| a | 1 | 0.0…0.3 | 0.160975225 |
| b | kg m$^{-3}$ °C$^{-1}$ | 0…20 | 7.41216035 |
| c | kg m$^{-3}$ | 100…250 | 218.46983092 |
| cps | 1 | 1.0…1.5 | 1.3065380539 |
| tm | °C | −1.0…2.0 | −0.4674846189 |
| pm | °C$^{-1}$ d$^{-1}$ | 0.0…1.0 | 0.4355929409 |
| ev | mm d$^{-1}$ | 0.0…0.2 | 0.0787463821 |
| kmax | mm °C$^{-1}$ d$^{-1}$ | 2.5…15.0 | 0.26291311*e$^{0.03958291*\lambda}$ |
| kmin | mm °C$^{-1}$ d$^{-1}$ | 0.1…2.5 | 1044.72422*e$^{-0.1025652*\lambda}$ |
| $\rho_{s,new_{min}}$ | kg m$^{-3}$ | 30…100 | 60.42091336 |
| $\rho_{s_{max}}$ | kg m$^{-3}$ | 200…400 | 291.42990453 |
| $\rho_{s,inc,rain}$ | kg m$^{-3}$ mm$^{-1}$ d$^{-1}$ | 0…10 | 5.40364768 |
| $\rho_{s,inc,age}$ | kg m$^{-3}$ d$^{-1}$ | 0…20 | 2.67193647 |
| $\rho_{s,inc,thaw}$ | kg m$^{-3}$ °C$^{-1}$ d$^{-1}$ | 0…20 | 6.22849401 |

[1]e=2.718281828459, λ=latitude in degrees north

**Table 5.** Interception coefficients, kmax coefficients and kmin coefficients used in this study for different forest types.

| Forest density class | Interception coefficient | Kmax coefficient | Kmin coefficient |
|---|---|---|---|
| Open area | 1.00 | 1.00 | 1.00 |
| Deciduous forest / sparse mixed forest | 0.92 | 0.65 | 0.875 |
| Pine forest | 0.86 | 0.60 | 0.85 |
| Dense spruce forest | 0.70 | 0.50 | 0.80 |

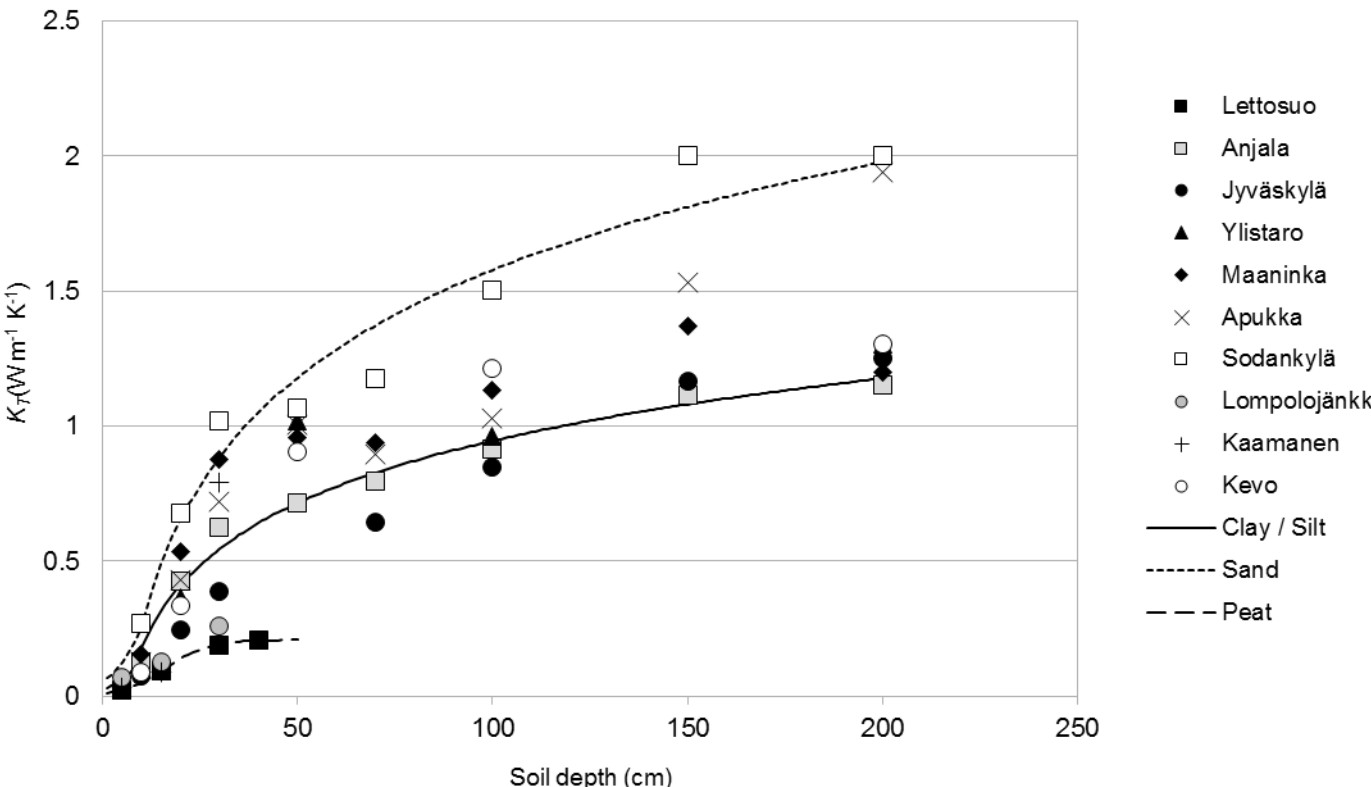

**Figure 1.** Calibrated $K_T$ values at each soil temperature measurement site and depth. Logistic regression curves fitted to the data from Anjala, Sodankylä and Lettosuo stations representing clay/silt, sand and peat soil types, respectively, are shown as well.

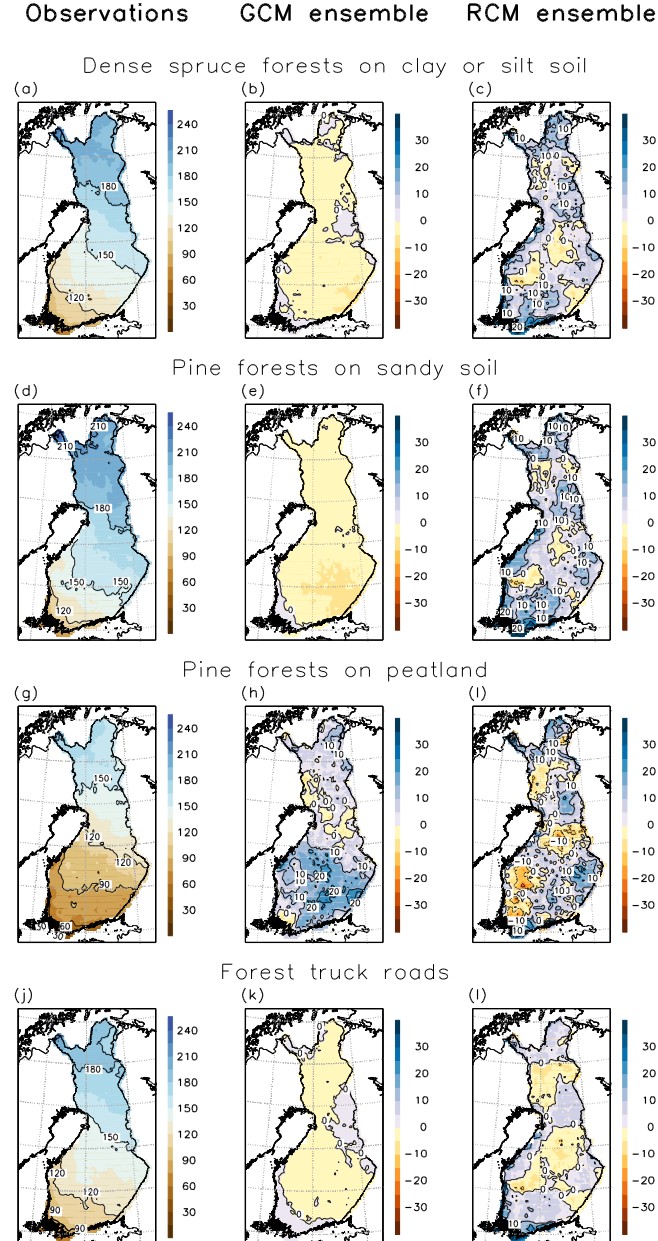

**Figure 2.** Annual average modelled number of days with good bearing capacity over the period 1981–2010 in three different forest and soil types and in forest truck roads based on observed weather data (left panel). The middle panel depicts the multi-model mean differences in the annual average number of days with good bearing capacity over the period 1981–2010 between the bias-corrected GCM ensemble and calculations using the observed weather data. In the right panel the same multi-model mean differences are showed for the bias-adjusted RCM ensemble.

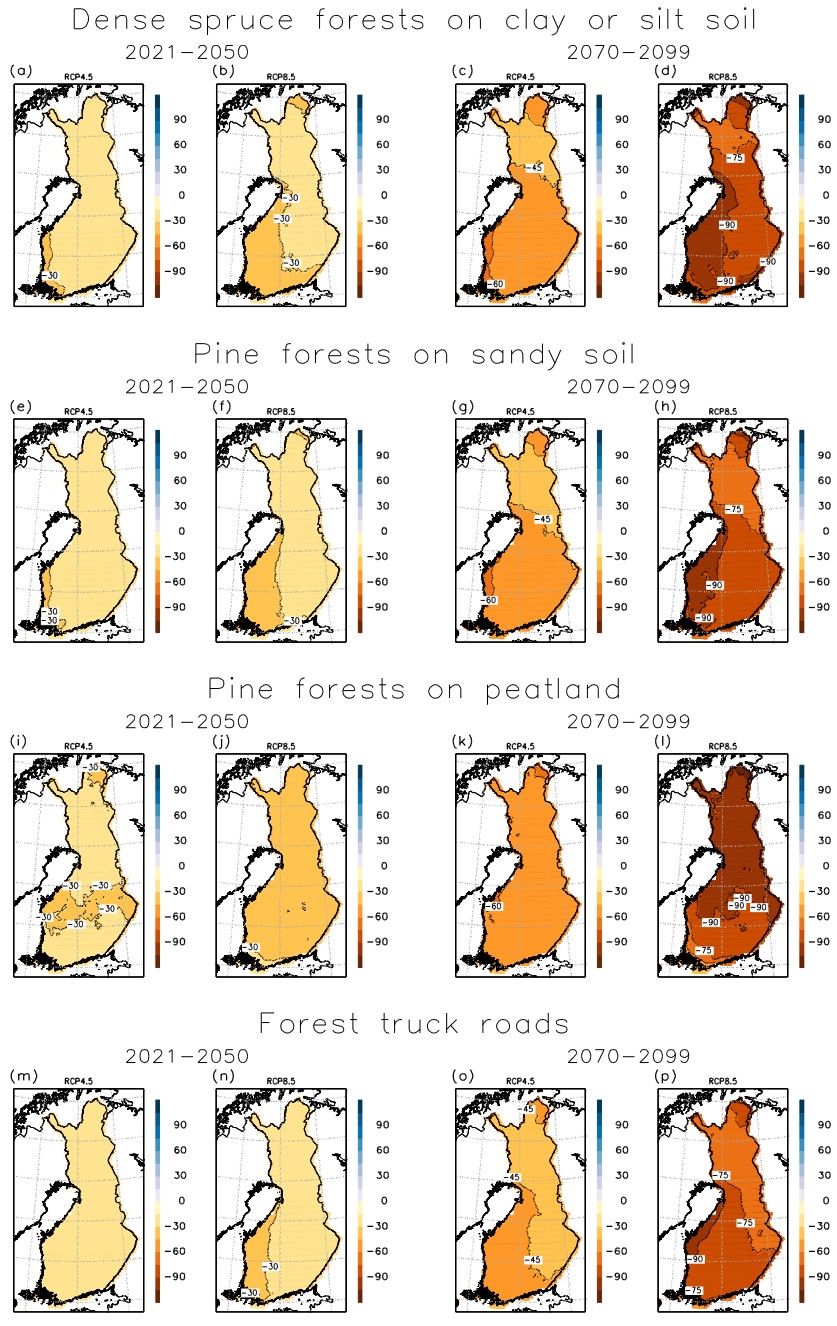

**Figure 3.** Projected multi-GCM mean change in the annual number of days with good bearing capacity in three different forest and soil types and in forest truck roads from 1981–2010 to 2021–2050 and to 2070–2099 under the RCP4.5 and RCP8.5 scenarios.

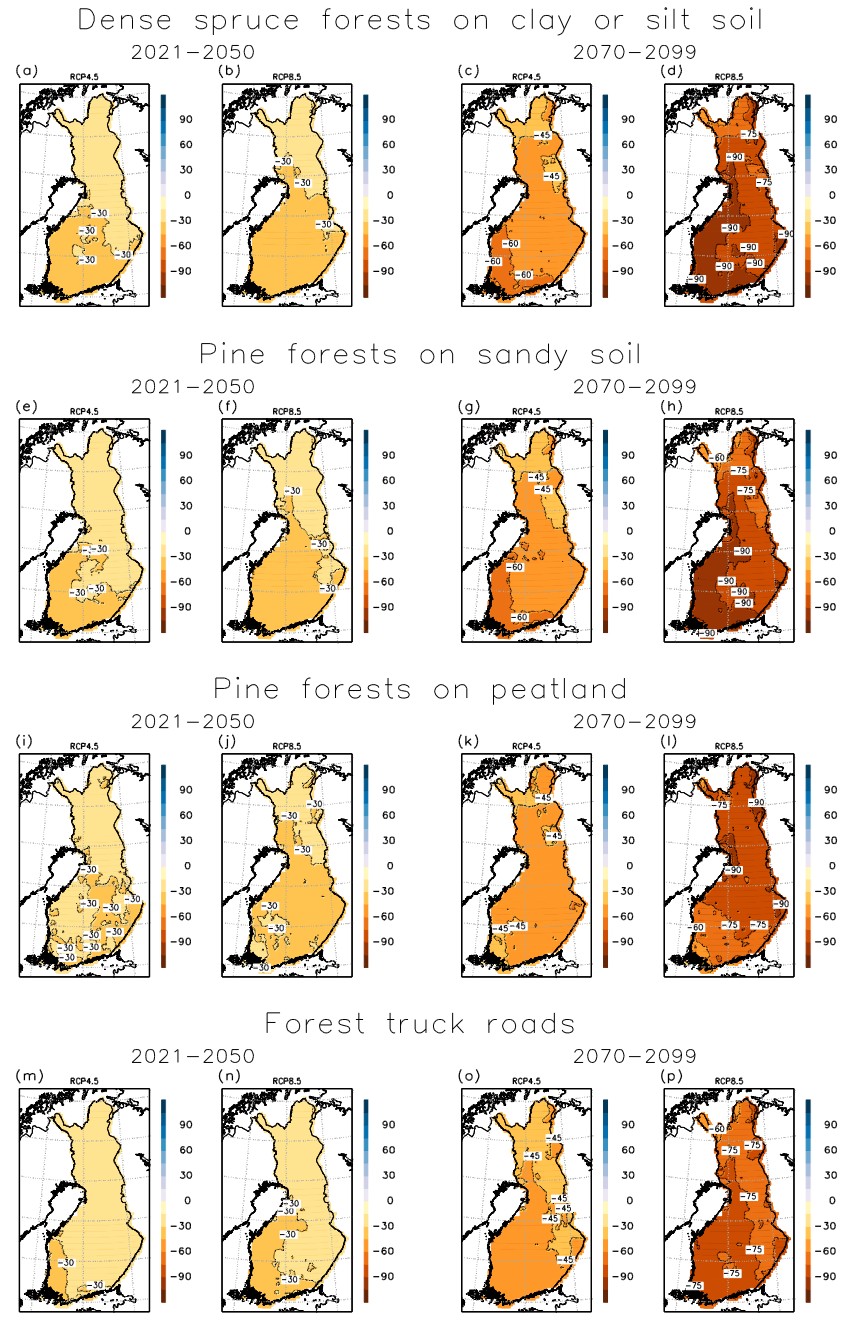

**Figure 4.** As in Fig. 3 but for multi-RCM mean change.

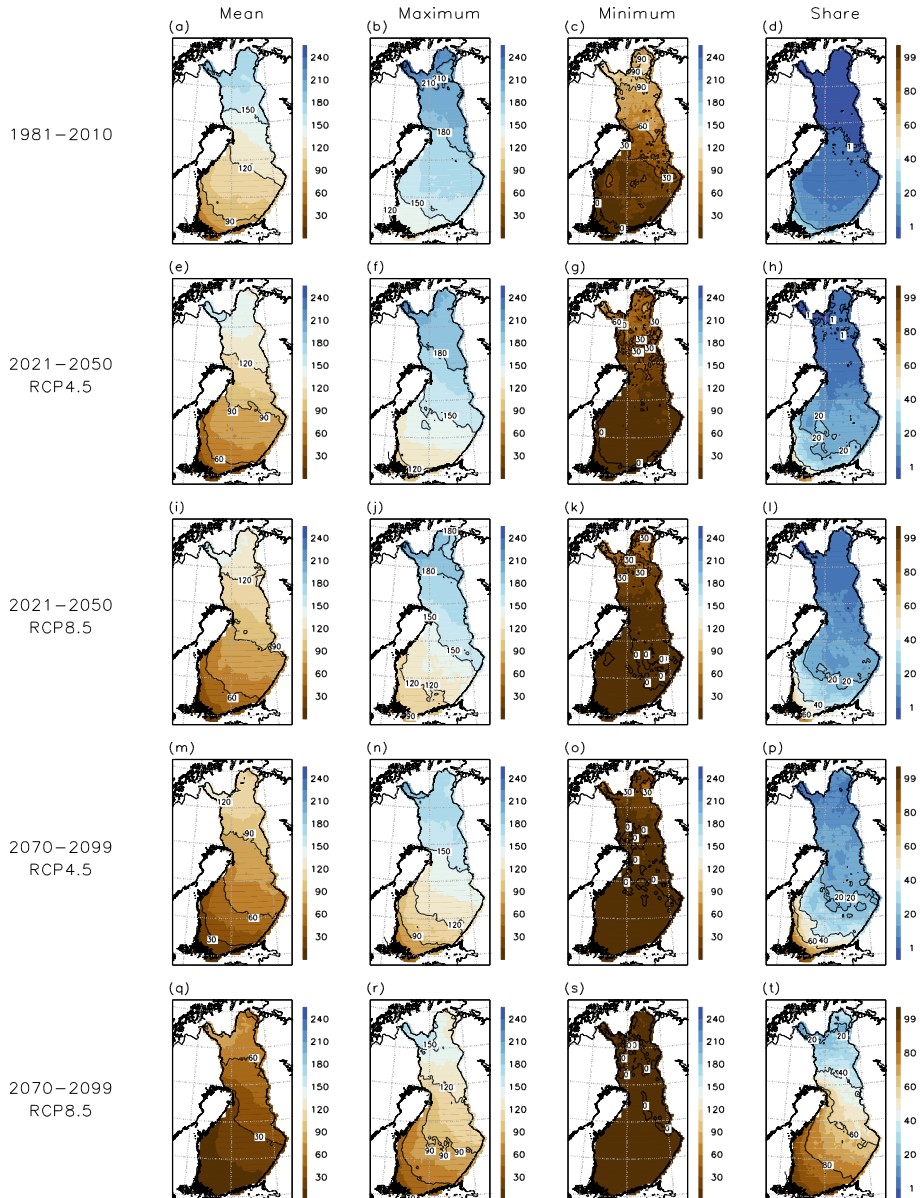

**Figure 5.** Modelled multi-GCM annual mean number of days with good bearing capacity in drained pine-dominated peatland forests during 1981–2010, 2021–2050 and 2070–2099 under the RCP4.5 and RCP8.5 scenarios (left panel). The second and third panel from the left show the modelled annual number of days with good bearing capacity during the winter with most (the second panel) and least (the third panel) such days within the 30-year periods based on the same multi-GCM mean. The last panel shows the share of winters (%) with less than 30 modelled days of good bearing capacity based on the multi-GCM mean.

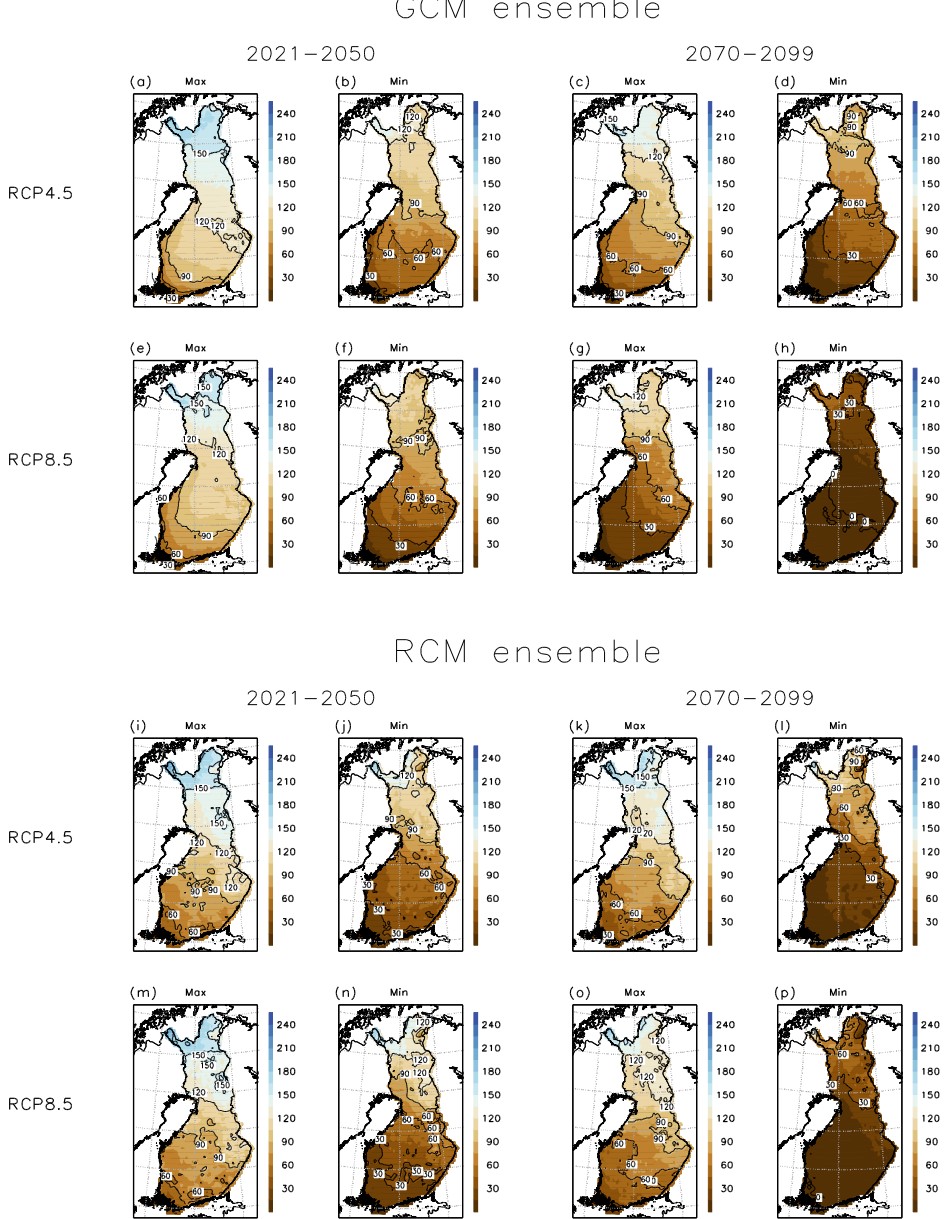

**Figure 6.** Range of modelled annual mean number of days with good bearing capacity in drained pine-dominated peatland forests during the periods 2021–2050 and 2070–2099 under the RCP4.5 and RCP8.5 scenarios among the GCMs and RCMs used in this study. The figures entitled with "Max" and "Min" show the highest and lowest modelled mean number of days with good bearing capacity among the used models for the GCMs and RCMs.

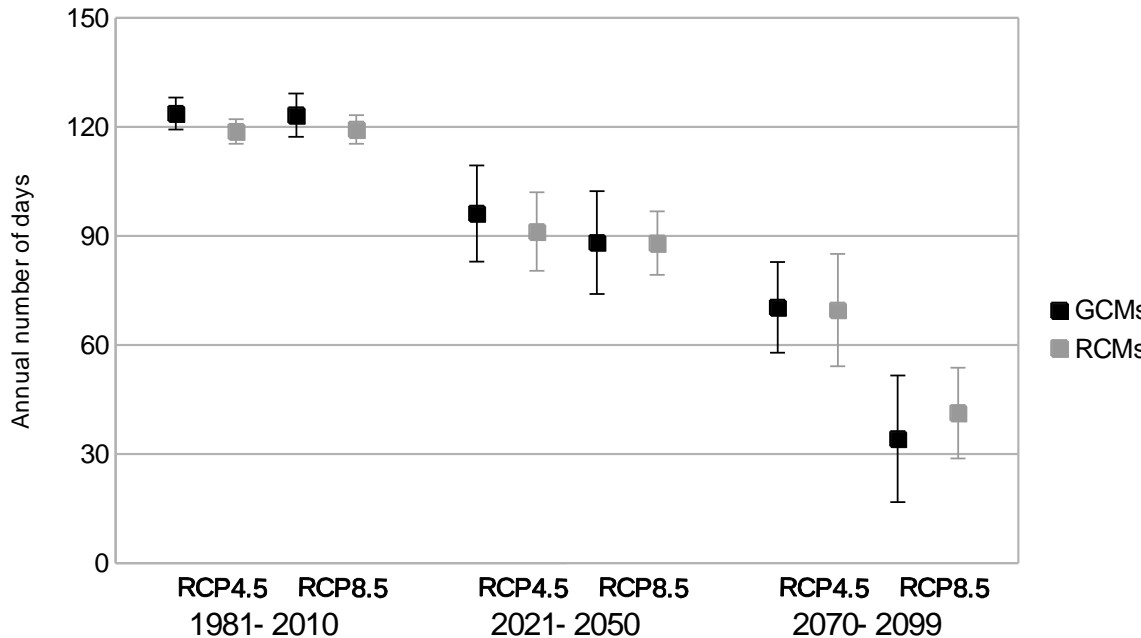

**Figure 7.** Annual mean number of days with good wintertime bearing capacity in drained pine-dominated peatland forests with multi-model standard deviations as averaged over whole of Finland separately for the GCM and RCM ensembles during the periods 1981–2010, 2021–2050 and 2070–2099 under the RCP4.5 and RCP8.5 scenarios.

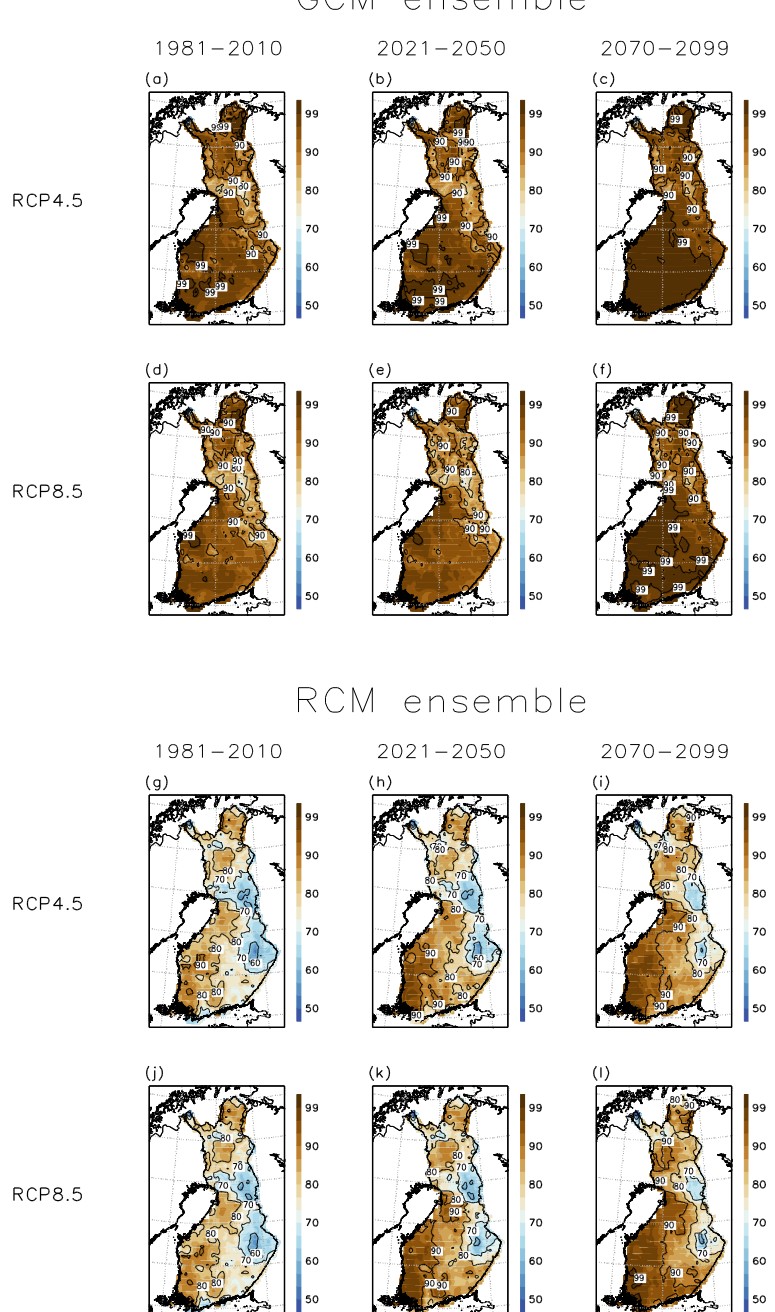

**Figure 8.** The share of modelled bearing season length (%) when the modelled soil frost depth exceeded 20 cm in drained pine-dominated peatland forests during the periods 1981–2010, 2021–2050 and 2070–2099 under the rcP4.5 and RCP8.5 scenarios among the GCMs and RCMs used in this study.