# Peer review of "Projected decrease in wintertime bearing capacity on different forest and soil types in Finland under a warming climate"

_Hydrology and Earth System Sciences, 2017_

## Referee Comment (RC1) · Anonymous Referee #1 · 2 Mar 2018

The paper presents the results of a study aimed to evaluate the projected decrease in the bearing capacity of Finnish soils in function of the changing climate during the 21st century. The paper appears well written and the results are interesting for the scientific community, even if related to the specific territory of Finland. The method is general and can be applied also in other nations in which the wood harvesting is economically important. However, a lack in this paper is the detail related to the choices of parameters performed in the model used, the description of the pre-processing procedures (inclusive of the choices of the several parameters used in this study), the statistical comment about the values (especially those selected as a result of several simulations), and in general a too short description about the consequences and the

limitations of these choices on the interpretation of the results. In my opinion, this part deserves a deepening, because it could help to evaluate the results and also give more strength and robustness to the conclusions. This is the reason for which I do not think that this paper could be accepted in the present form, but requires some modifications that, in my view, can be intended as minor. The list of requirements can be understood better by looking at the specific comments here listed page by page (see supplement file).

Please also note the supplement to this comment:
https://www.hydrol-earth-syst-sci-discuss.net/hess-2017-727/hess-2017-727-RC1-supplement.pdf

**Supplement:**

**Review of paper "Projected decrease in wintertime bearing capacity on different forest and soil types in Finland under a warming climate" by Lehtonen et al.**

The paper presents the results of a study aimed to evaluate the projected decrease in the bearing capacity of Finnish soils in function of the changing climate during the 21st century. The paper appears well written and the results are interesting for the scientific community, even if related to the specific territory of Finland. The method is general and can be applied also in other nations in which the wood harvesting is economically important.

However, a lack in this paper is the detail related to the choices of parameters performed in the model used, the description of the pre-processing procedures (inclusive of the choices of the several parameters used in this study), the statistical comment about the values (especially those selected as a result of several simulations), and in general a too short description about the consequences and the limitations of these choices on the interpretation of the results. In my opinion, this part deserves a deepening, because it could help to evaluate the results and also give more strength and robustness to the conclusions. This is the reason for which I do not think that this paper could be accepted in the present form, but requires some modifications that, in my view, can be intended as minor. The list of requirements can be understood better by looking at the specific comments here listed page by page.

- Introduction: in my opinion, a too large part of the introduction is dedicated to explain the industrial problems, while a too small part is dedicated to the scientific problem and the models used.
- Page 4 lines 4-28: the equation proposed to estimate soil temperature seems not consider the effects of soil moisture (unless thermal conductivity is kept variable, but since there are no measures of soil moisture it is hard to consider such variations). A comment on this consideration may be required.
- Page 5 line 15: regarding KT values, is the interval of values used significant for the considered soils?
- Page 5 lines 18-19: "while, for example, KT seemed to steadily increase with soil depth." this is consistent with the assumption of increasing soil moisture at increasing depth (or change of soil texture): do you have any data evidencing these facts? Please comment.
- Pages 4-5: the method elaborated to retrieve soil thermal conductivity is strongly linked to the availability of soil temperature data, and thus will become representative of the experimental sites during the measurement periods. If I have correctly understood, such values optimized for each site will be adopted for the following simulations. However, there is no any reason for which such values could remain constant also in future climate... This could be a limitation for the reliability of future simulations. If authors do not agree with my conclusion, they could explain why...
- Page 6 lines 1-10: the choice of different thresholds for soil freezing changed substantially the evaluation of the number of days with frozen soil? How and how much?
- Page 6 lines 17-23, and page 7 line 5: I suggest to say here that the values used in eqs. 5 and 6 will be discussed later.
- Page 8 lines 3-12: again, the method elaborated to retrieve the values of parameters is strongly linked to the availability of measured data, and thus will become representative of the experimental sites during the measurement periods. Since such values optimized for each site will be adopted for the following simulations, there is no any reason for which such values could remain constant also in future climate... Also in this case, if authors have a different idea, they could explain why...
- Pages 8-9 lines 29-4: in this paper, many decisions about parameters are just summarized by "hiding" the results. For instance, in this case, the choice of values for kmin and kmax is not

justified, and the reader cannot understand how it has been made. In my opinion, this may deserve an additional subsection (similarly as all other choices of this model).

- Page 9 lines 9-10: authors use only R2 as indicator of good simulations. However, if just for example I would have a simulation in which simulated snow depth has almost the same time trend of observations, but a value that is double, R2 will be close to one even if the relative error will be 200%... I suggest to use also bias or standard error as a criterion to validate simulations, and not only use correlation coefficient (and, by the way, it is better to use R and not R2).
- Page 9 lines 30-32: how the GCM and RCM have been chosen (I think you should mention here more clearly that the detailed list of model chosen is reported in Table S3), and why those models, among the whole EURO-CORDEX dataset?
- Page 10 line 20: is the modeled annual average number of days evaluated as the average of all GCMs and RCMs, respectively?
- Page 12 lines 6-15: how large is the difference among model ensembles (separately for RCM and GCM) in the three climatic periods? I think that also this information is important to statistically locate your results. Section 3,4 and Figure 6, in my opinion, are not informative, as they mention only the two models giving the maximum and the minimum values, and not the distribution. As climate cannot be described just by extremes, but needs a complete statistical information, for the same reason I think that the standard deviation or some equivalent statistical parameter can be more informative about the dispersion of individual model calculations.
- Table 3: the numerical values given for each parameter have too many digits, most of them without any statistical meaning. Instead of giving a number with too many not significant digits, authors should give a number and an error associated with the experiments and comparisons, like a = value ± error
- Figure 2: since it is hard to appreciate differences among the three figures, given the quite large interval of variation of the number of days, it could be better to plot, for second and third column, the differences among GCM and observations, and RCM and observations, respectively (similarly to what you did for Fig, 3). Or maybe you can add such figures, if you want to keep the total number of days.

---

## Referee Comment (RC2) · Anonymous Referee #2 · 31 Mar 2018

**Review of paper "Projected decrease in wintertime bearing capacity on different forest and soil types in Finland under a warming climate" by Lehtonen et al.**

Wood harvesting is an important part of the Finnish economy. The current national strategy (2014-2020) is also to increase the economic output from this industry, and over the next decades increasing demand is expected to put further pressure on the annual wood harvesting. This has traditionally been a winter activity when the ground is frozen and the mobility is greater and impacts on the natural areas are smaller. The paper aims to evaluate the projected decrease in the bearing capacity of Finnish soils, especially dried peatland, under two different projected warming scenarios for the 21st century. The wood harvesting should preferably be undertaken (and increased) throughout the year, due to a steady demand by the processing industry. The results are presented across Finland for different combinations of three soil types and four forest types, in addition to forest truck roads. Hence, the results take into account that for a given location, the soil and/or forest might change in the future. The relevance of the study is therefore clear but the potential or expected impact could be better explained (in the context of year-round harvesting with new methods and machinery in a changing climate) also because this study should have interest outside Finland and Finnish interests. The abstract provides a concise and complete summary, with the exception of not mentioning the snow model and using vague terms like "largely determined by" and "mainly determined by". Overall the paper is well structured and clear but could be shortened. The language is generally good with a few exceptions which will be fixed by a critical shortening of the text.

The bearing capacity of frozen soil is an important parameter is this study. The authors claim, with reference to Eeronheimo (1991), that a 20 cm thick layer of frozen soil or 40 cm thick layer of ground snow can bear standard (heavy) vehicles (15-30 tons) used in forest harvesting. Eeronheimo is written in Finnish with an English summary at the end where it is stated: "The logging conditions in peatland forests are often unfavourable. The bearing capacity of the ground is poor most of the year and difficult to determine, which complicates felling and extraction practices. According to forest harvesting specialists there should be either frost layer of at least 20 cm or, when there is no frost, snow cover of 40 cm or more to facilitate extraction with medium-sized forwarders." Using published guidelines, Shoop (1995) proposed a relationship for bearing capacity of frozen ground, including as a function of dry and wet soil conditions. Dry conditions (dried peatlands are abundant in Finland) require a deeper freezing layer and a 10-ton truck needs 0.35-0.50 m frozen soil thickness, according to Table 1 in Shoop (1995), where the upper limit (0.50 m) is said to be a conservative estimate (page 555). (In the list of studies citing Shoop there are also more recent studies although they do not seem to bring any significant new knowledge on bearing capacity of frozen soil). In addition the distribution of the ground pressure (e.g. wheel load) from the vehicles/machines may impact the bearing capacity (breakthrough failure versus localized crushing). The relationship between the frozen soil layer and vehicle/machine bearing-capacity therefore seems more complex than put forward by the authors in this study. At least soil freezing depth and bearing capacity requires a more in-depth discussion (explaining also what is meant by "idealized approach" on page 13 line 8), but possibly a new analysis of the results. The claim that 40 cm thick ground snow (over frost free soil) ensures the

same bearing capacity as 20 cm frozen soil should be supported by more evidence than from Eeronheimo. For instance, is this regardless of snow density? I guess not.

It is correct that there are "several models designed for calculation of soil temperatures" (page 3 line 14) but the authors list only a few. For instance SURFEX (including different snow pack models like Crocus), see Special Issue in GMD at https://www.geosci-model-dev.net/special_issue14.html), where FMI has experience and expertise. There are most likely good reasons for the author's choice of land surface model but they are not well presented (and discussed) in the paper. Moreover, the employed GCMs and RCMs use land surface models and it should be explained why direct use of these results for soil temperature, or stand-alone high-resolution implementations, are not presented or used. This also applies to snow depth. I can speculate about the answer but not all readers might be well enough into surface modelling to do so. More details are also relevant for the validation of the modelled temperatures and snow depth. For soil temperatures the validation of the optimized model shows that either correlation or the number of days with soil temperatures below freezing can be well represented (but what is meant by "greatly overestimated" or "dramatically worse", top of page 6?). The results (correlation and number of freezing days) after introducing soil freezing points below zero are also not presented. These below zero freezing points are based on a study in Finnish and therefore difficult to use as reference – are there other studies which are more appropriate to use?

Another reference should also be used for the determination of precipitation phase. Also, it should be explained why this classification is needed as not all readers are familiar with the (limited) output from numerical weather and climate simulations.

About the snow model, it is based largely on another study. From the text it is not very clear what is new and what is taken from the existing model. Can the text be shortened accordingly?

Is the modelled snow depths validated for RMSE? Or for the snow depth threshold of 40 cm?

The calibration periods for soil temperature and snow depth are relatively short. Is there a risk that they are too short when the resulting models are used on future climate conditions, cf. last sentence in Section 2.2.3?

The study "modelled the number of days with good bearing capacity in the forest harvesting point of view. (…) soil frost (…) at least the depth of 20 cm or when the snow depth exceeded 40 cm" (page 9, lines 27-29). I think it would add to the study to present the bearing capacity separately for soil frost thickness and snow depth. Both variables are dependent on climate change but their sensitivities to changes might be different, e.g. freezing temperature before or after snowfall affects the soil freezing layer differently. Also, such a separation should be beneficial both for the validity of the used model and when projected values are compared to current conditions (baseline period).

The evaluation of the methodology (Section 4.2) should be extended, following the suggestions above. Also, the comparison to Eeronheimo (and his Fig. 4) – page 13, lines 9-12) – could be done more thoroughly, perhaps using a map of soil bearing capacity which reflects the actual soil and forest properties in each grid point.

Are the GCM and RCM ensemble means used in Fig. 2?

I might be wrong, but it seems that in several of the panels in Figs. 2-6 the bearing capacity also extends over sea. I am guessing this is related to model resolution and the many Finish islands. But are these of interest to this study or can they be omitted from the presentation for (this readers) improved readability?

Shoop, S. A., Vehicle bearing capacity of frozen ground over a soft substrate. Canadian Geotechnical Journal, 1995, Vol. 32, No. 3 : pp. 552-556. https://doi.org/10.1139/t95-057 (http://www.nrcresearchpress.com/doi/abs/10.1139/t95-057#.Wr9Lki5uapo)

---

## Author Comment (AC1) · 27 Apr 2018

**Journal:** Hydrology and Earth System Sciences
**Title:** Projected decrease in wintertime bearing capacity on different forest and soil types in Finland under a warming climate
**Authors:** I. Lehtonen, A. Venäläinen, M. Kämäräinen, A. Asikainen, J. Laitila, P. Anttila and H. Peltola
**MS No.:** hess-2017-727
**MS Type:** Research Article
**Iteration:** First review
**Referee #1**

*We would like to thank the referee for the constructive comments and suggestions. Our replies to the comments are given in "Italics" after each specific comment.*

The paper presents the results of a study aimed to evaluate the projected decrease in the bearing capacity of Finnish soils in function of the changing climate during the 21$^{st}$ century. The paper appears well written and the results are interesting for the scientific community, even if related to the specific territory of Finland. The method is general and can be applied also in other nations in which the wood harvesting is economically important.
However, a lack in this paper is the detail related to the choices of parameters performed in the model used, the description of the pre-processing procedures (inclusive of the choices of the several parameters used in this study), the statistical comment about the values (especially those selected as a result of several simulations), and in general a too short description about the consequences and the limitations of these choices on the interpretation of the results. In my opinion, this part deserves a deepening, because it could help to evaluate the results and also give more strength and robustness to the conclusions. This is the reason for which I do not think that this paper could be accepted in the present form, but requires some modifications that, in my view, can be intended as minor. The list of requirements can be understood better by looking at the specific comments here listed page by page.

− Introduction: in my opinion, a too large part of the introduction is dedicated to explain the industrial problems, while a too small part is dedicated to the scientific problem and the models used.

*We agree that the introduction can be reorganized and additionally shortened in general.*

− Page 4 lines 4-28: the equation proposed to estimate soil temperature seems not consider the effects of soil moisture (unless thermal conductivity is kept variable, but since there are no measures of soil moisture it is hard to consider such variations). A comment on this consideration may be required.

*Yes, the equation assumes constant water content over time. We agree that it would be a good idea to add a comment about this in the manuscript.*

− Page 5 line 15: regarding KT values, is the interval of values used significant for the considered soils?

*Yes, we think so. According to the study by Rankinen et al. (2004) where the used soil temperature model was first introduced, the calibrated $K_T$ values varied between 0.5 and 0.8 at five stations across Finland. They had calibrated $K_T$ at 20 cm and 50 cm depths and the highest value they got (~0.8) was*

*at Sodankylä station, which seemed to have the highest $K_T$ values also among our stations. The soil type at Sodankylä is sandy gravel and soil types with more fine-grained texture tend to have smaller $K_T$ values. Jungqvist et al. (2014) used optimization interval $0...1\ Wm^{-1}K^{-1}$ for $K_T$ but we noted that $1\ Wm^{-1}K^{-1}$ is not enough for the higher limit at deeper soil depths so we extended the higher limit to $2\ Wm^{-1}K^{-1}$ although we mainly focused on 20 cm soil depth and at that depth, the optimized values did not exceed 0.7.*

− Page 5 lines 18-19: "while, for example, KT seemed to steadily increase with soil depth." this is consistent with the assumption of increasing soil moisture at increasing depth (or change of soil texture): do you have any data evidencing these facts? Please comment.

*This is undoubtedly one reason for this. For example, in the report by Soveri and Varjo (1977) cited in the manuscript, they show in Table 4 measured soil moistures from one test site over one winter season. Based on those observations, the soil moisture, on average, increases with increasing soil depth. In Table 5 they show typical heat capacity and heat conductivity values for different soil types with different soil moistures. Based on those values, the heat conductivity increases rapidly with soil moisture. In the report by Heikinheimo and Fougstedt (1992) are shown the soil textures at some of the stations on certain depths. For example, at Anjala the share of clay increases with increasing depth whereas, e.g., at Maaninka there is almost equal amount of silt and sand near the surface and also at 0.7 m depth but mainly sand around 0.5 m depth and below 0.7 m.*

− Pages 4-5: the method elaborated to retrieve soil thermal conductivity is strongly linked to the availability of soil temperature data, and thus will become representative of the experimental sites during the measurement periods. If I have correctly understood, such values optimized for each site will be adopted for the following simulations. However, there is no any reason for which such values could remain constant also in future climate... This could be a limitation for the reliability of future simulations. If authors do not agree with my conclusion, they could explain why...

*Yes, the parameter values were kept constant throughout the simulations after the parameters were defined for the three soil types based on the calibration period. Of course, it is clear that there are a lot of uncertainty in the parameter values and in reality they are never absolutely equal in two different places. Moreover, an almost equally good correlation between the observed and simulated soil temperatures can be achieved with very different set of parameters: if you modify one parameter, you can further modify the rest of parameters conveniently to achieve virtually still as high correlation as previously. Thus, we first set values for the least sensitive parameters and only $K_T$ was optimized during the last phase when all the other parameters were kept constant as the results seemed to be most sensitive to $K_T$. At this stage, we moreover used only one station representing one soil type, so these three stations are basically representative examples for the soil types. However, as can be seen from Table S1, high $R^2$ values were achieved also at many other stations for different soil types, e.g., at Maaninka both for clay/silt and sandy soil. The model performed reasonably well even with the wrong soil type at many locations during the calibration period (see Table S1). As the used stations are moreover located in areas representing quite a different climatic conditions, we assume that possible changes in soil characteristics, including thermal conductivity, do not crucially change the results.*

− Page 6 lines 1-10: the choice of different thresholds for soil freezing changed substantially the evaluation of the number of days with frozen soil? How and how much?

*For example, at Ylistaro station the soil temperature was during the calibration period below 0 degC at 20 cm depth on 81 days and at 50 cm depth on 58 days annually, on average, when taking into account that soil temperature was observed once every fifth day. The modelled soil temperature was below 0 degC for clay/silt soil type configuration at 20 cm depth on 132 days and at 50 cm depth on 124 days annually. The modelled soil temperature was below -0.5 degC at 20 cm depth on 103 days and at 50 cm depth on 51 days annually. In addition, the observed soil temperature at 20 cm depth was also on 30 days annually exactly at 0 degC as the accuracy of soil temperature observations was 0.1 degC.*

*There was also huge variability in the number of days with soil temperatures below 0 degC between individual stations, evidently induced by differences in local characteristics. For example, at Apukka station soil temperature at 20 cm was only on 35 days annually below 0 degC while at Sodankylä there were 178 such days and at Lompolojänkkä the observed soil temperature had never (only three years of observations from there) been below 0 degC at any measurement depth. All of these three stations are located in Lapland between 66th and 68th northern latitude in similar climatic conditions. These three stations had very different soil characteristics but also within stations with more similar soil types there existed surprisingly notable variability in the number of frozen days.*

– Page 6 lines 17-23, and page 7 line 5: I suggest to say here that the values used in eqs. 5 and 6 will be discussed later.

*Ok*

– Page 8 lines 3-12: again, the method elaborated to retrieve the values of parameters is strongly linked to the availability of measured data, and thus will become representative of the experimental sites during the measurement periods. Since such values optimized for each site will be adopted for the following simulations, there is no any reason for which such values could remain constant also in future climate... Also in this case, if authors have a different idea, they could explain why...

*The parameters optimized at each station located in different climatic conditions across Finland were averaged over all the stations to achieve the final parameters, which were then used in validation of the snow model (except kmax and kmin related to the solar azimuth angle having the latitudinal dependence). The validation period moreover had different kind of winters, cold and mild, snowy etc. If we had used the parameter values optimized for each station, the $R^2$-values would have been approximately 0.01 higher (page 8, line 15). Based on Table S2, the snow model performed equally well in different climatic conditions in different parts of Finland. However, we clearly see that the model with the optimized parameters performed worse before 1981 than thereafter, which we think is largely attributed to the correction factor for solid precipitation (cps), which we think had been higher previously due to a larger measurement error. Of course, it is not impossible that there have been some shifting in other parameters as well. On the other hand, part of the parameters are linked to things like freezing point of water or solar azimuth angle, which we can easily assume to stay constant.*

– Pages 8-9 lines 29-4: in this paper, many decisions about parameters are just summarized by "hiding" the results. For instance, in this case, the choice of values for kmin and kmax is not justified, and the reader cannot understand how it has been made. In my opinion, this may deserve an additional subsection (similarly as all other choices of this model).

*We will elaborate the choice of the parameters in more detail. The impact of forest canopy for kmin and kmax was estimated based on Vehviläinen (1992).*

– Page 9 lines 9-10: authors use only R2 as indicator of good simulations. However, if – just for example - I would have a simulation in which simulated snow depth has almost the same time trend of observations, but a value that is double, R2 will be close to one even if the relative error will be 200%... I suggest to use also bias or standard error as a criterion to validate simulations, and not only use correlation coefficient (and, by the way, it is better to use R and not R2).

*Due to this issue, in calibrating the snow model we minimized root mean square error (p. 8, l. 8) instead of maximizing $R^2$. In Table S2 we show in addition to $R^2$ the relative error. During the calibration period, the modelled and observed snow depths are, on average, close to each other in addition to high $R^2$.*

– Page 9 lines 30-32: how the GCM and RCM have been chosen (I think you should mention here more clearly that the detailed list of model chosen is reported in Table S3), and why those models, among the whole EURO-CORDEX dataset?

*We will add details about the model choice. Basically, these GCMs were chose because we had done the bias-correction for those models in a previous project. Those models were originally chose based on their skill in simulating present-day temperature and precipitation climatology over northern Europe. The RCMs were chose because we wanted to use a set of models with a uniform bias-adjustment approach and this set of models with a uniform bias-adjustment approach had a largest number of simulations available.*

– Page 10 line 20: is the modeled annual average number of days evaluated as the average of all GCMs and RCMs, respectively?

*Yes, the multi-model means are shown in the figure.*

– Page 12 lines 6-15: how large is the difference among model ensembles (separately for RCM and GCM) in the three climatic periods? I think that also this information is important to statistically locate your results. Section 3,4 and Figure 6, in my opinion, are not informative, as they mention only the two models giving the maximum and the minimum values, and not the distribution. As climate cannot be described just by extremes, but needs a complete statistical information, for the same reason I think that the standard deviation or some equivalent statistical parameter can be more informative about the dispersion of individual model calculations.

*We partly agree and partly disagree with this comment. We think that Fig. 6 showing the complete spread among the model ensembles is informative but of course some more information about the distribution could be also added. For example, a scatter plot showing the area-averaged number of days with good bearing capacity for each individual model would illustrate the distribution very well.*

− Table 3: the numerical values given for each parameter have too many digits, most of them without any statistical meaning. Instead of giving a number with too many not significant digits, authors should give a number and an error associated with the experiments and comparisons, like a = value ± error

*It is hard to estimate the error associated with most of the parameters as in many cases if you change the value of one parameter completely, the model can be still adjusted to perform equally well by adjusting the values of other parameters as well. So, the parameters are intrinsically not exact at all but we have given here those values that were used in our calculations. Moreover, for example, one of the parameters is Napier's constant and we are unsure how to measure the error in its value.*

− Figure 2: since it is hard to appreciate differences among the three figures, given the quite large interval of variation of the number of days, it could be better to plot, for second and third column, the differences among GCM and observations, and RCM and observations, respectively (similarly to what you did for Fig, 3). Or maybe you can add such figures, if you want to keep the total number of days.

*This can be changed.*

---

## Author Comment (AC2) · 27 Apr 2018

**Journal:** Hydrology and Earth System Sciences
**Title:** Projected decrease in wintertime bearing capacity on different forest and soil types in Finland under a warming climate
**Authors:** I. Lehtonen, A. Venäläinen, M. Kämäräinen, A. Asikainen, J. Laitila, P. Anttila and H. Peltola
**MS No.:** hess-2017-727
**MS Type:** Research Article
**Iteration:** First review
**Referee #2**

*We would like to thank the referee for the comments and suggestions. Our replies to the comments are given in "Italics" after the comments given in the beginning of this document.*

Wood harvesting is an important part of the Finnish economy. The current national strategy (2014-2020) is also to increase the economic output from this industry, and over the next decades increasing demand is expected to put further pressure on the annual wood harvesting. This has traditionally been a winter activity when the ground is frozen and the mobility is greater and impacts on the natural areas are smaller. The paper aims to evaluate the projected decrease in the bearing capacity of Finnish soils, especially dried peatland, under two different projected warming scenarios for the 21st century. The wood harvesting should preferably be undertaken (and increased) throughout the year, due to a steady demand by the processing industry. The results are presented across Finland for different combinations of three soil types and four forest types, in addition to forest truck roads. Hence, the results take into account that for a given location, the soil and/or forest might change in the future. The relevance of the study is therefore clear but the potential or expected impact could be better explained (in the context of year-round harvesting with new methods and machinery in a changing climate) also because this study should have interest outside Finland and Finnish interests. The abstract provides a concise and complete summary, with the exception of not mentioning the snow model and using vague terms like "largely determined by" and "mainly determined by". Overall the paper is well structured and clear but could be shortened. The language is generally good with a few exceptions which will be fixed by a critical shortening of the text.

*We agree with these remarks by the reviewer.*

The bearing capacity of frozen soil is an important parameter is this study. The authors claim, with reference to Eeronheimo (1991), that a 20 cm thick layer of frozen soil or 40 cm thick layer of ground snow can bear standard (heavy) vehicles (15-30 tons) used in forest harvesting. Eeronheimo is written in Finnish with an English summary at the end where it is stated: "The logging conditions in peatland forests are often unfavourable. The bearing capacity of the ground is poor most of the year and difficult to determine, which complicates felling and extraction practices. According to forest harvesting specialists there should be either frost layer of at least 20 cm or, when there is no frost, snow cover of 40 cm or more to facilitate extraction with medium-sized forwarders." Using published guidelines, Shoop (1995) proposed a relationship for bearing capacity of frozen ground, including as a function of dry and wet soil conditions. Dry conditions (dried peatlands are abundant in Finland) require a deeper freezing layer and a 10-ton truck needs 0.35-0.50 m frozen soil thickness, according to Table 1 in Shoop (1995), where the upper limit (0.50 m) is said to be a conservative estimate (page 555). (In the list of studies citing Shoop there are also more recent studies although they do not seem to bring any

significant new knowledge on bearing capacity of frozen soil). In addition the distribution of the ground pressure (e.g. wheel load) from the vehicles/machines may impact the bearing capacity (breakthrough failure versus localized crushing). The relationship between the frozen soil layer and vehicle/machine bearing-capacity therefore seems more complex than put forward by the authors in this study. At least soil freezing depth and bearing capacity requires a more in-depth discussion (explaining also what is meant by "idealized approach" on page 13 line 8), but possibly a new analysis of the results. The claim that 40 cm thick ground snow (over frost free soil) ensures the same bearing capacity as 20 cm frozen soil should be supported by more evidence than from Eeronheimo. For instance, is this regardless of snow density? I guess not.

*We agree that soil moisture affects to the critical soil frost depth. However, dried peatlands are not "dry" soils literally speaking, they are rather the wettest sites where some kind of forest grows in Finland. We can add more discussion considering the chosen thresholds, e.g., based on the study by Suvinen (2006).*

*With the "idealized approach" we mean that we use these certain classes for soil types and forest types but in reality, there are much more variability in soil conditions, so the inspected conditions are idealized in that sense that exactly the same kind of conditions probably occur nowhere.*

It is correct that there are "several models designed for calculation of soil temperatures" (page 3 line 14) but the authors list only a few. For instance SURFEX (including different snow pack models like Crocus), see Special Issue in GMD at https://www.geosci-model-dev.net/special_issue14.html), where FMI has experience and expertise. There are most likely good reasons for the author's choice of land surface model but they are not well presented (and discussed) in the paper. Moreover, the employed GCMs and RCMs use land surface models and it should be explained why direct use of these results for soil temperature, or stand-alone high-resolution implementations, are not presented or used. This also applies to snow depth. I can speculate about the answer but not all readers might be well enough into surface modelling to do so. More details are also relevant for the validation of the modelled temperatures and snow depth. For soil temperatures the validation of the optimized model shows that either correlation or the number of days with soil temperatures below freezing can be well represented (but what is meant by "greatly overestimated" or "dramatically worse", top of page 6?). The results (correlation and number of freezing days) after introducing soil freezing points below zero are also not presented. These below zero freezing points are based on a study in Finnish and therefore difficult to use as reference – are there other studies which are more appropriate to use?

*The climate models themselves tend to have too much soil frost but as we used bias-adjusted simulation data, the use of soil temperature or snow depth data directly from the models was not even possible as the similar bias-adjustment approaches could not be meaningfully applied to these data. With our own model we were moreover able to calculate the results for desired combinations of soil and forest types. Furthermore, we will pay more attention to the points mentioned by the referee.*

Another reference should also be used for the determination of precipitation phase. Also, it should be explained why this classification is needed as not all readers are familiar with the (limited) output from numerical weather and climate simulations.

*We can remove the reference to Hankimo (1976) and refer only to Vehviläinen (1992) and moreover add a short explanation why the classification is needed.*

About the snow model, it is based largely on another study. From the text it is not very clear what is new and what is taken from the existing model. Can the text be shortened accordingly?

*Vehviläinen (1992) describes different approaches that can be used in snow cover modelling and we have took relevant pieces from these approaches. As Vehviläinen (1992) concentrates mainly on snow water equivalent, we have added also features describing the density of snow cover.*

Is the modelled snow depths validated for RMSE? Or for the snow depth threshold of 40 cm?

*The snow model was optimized based on RMSE. $R^2$ and relative error are used to describe the validity of the model.*

The calibration periods for soil temperature and snow depth are relatively short. Is there a risk that they are too short when the resulting models are used on future climate conditions, cf. last sentence in Section 2.2.3?

*We suspect that the issue mentioned in the end of the section 2.2.3 is rather related to changes in precipitation measurements than an indication of different physical relationship between the weather and snow conditions. Of course, this possibility cannot be excluded that the calibration period may be too short.*

The study "modelled the number of days with good bearing capacity in the forest harvesting point of view. (...) soil frost (...) at least the depth of 20 cm or when the snow depth exceeded 40 cm" (page 9, lines 27-29). I think it would add to the study to present the bearing capacity separately for soil frost thickness and snow depth. Both variables are dependent on climate change but their sensitivities to changes might be different, e.g. freezing temperature before or after snowfall affects the soil freezing layer differently. Also, such a separation should be beneficial both for the validity of the used model and when projected values are compared to current conditions (baseline period).

*This kind of comparison can be added.*

The evaluation of the methodology (Section 4.2) should be extended, following the suggestions above. Also, the comparison to Eeronheimo (and his Fig. 4) – page 13, lines 9-12) – could be done more thoroughly, perhaps using a map of soil bearing capacity which reflects the actual soil and forest properties in each grid point.

*We will take these points into consideration. Many grid points have all kind of terrains because the size of individual grid points is close 100 km².*

Are the GCM and RCM ensemble means used in Fig. 2?

*Yes, the multi-model means are shown in the figure.*

I might be wrong, but it seems that in several of the panels in Figs. 2-6 the bearing capacity also extends over sea. I am guessing this is related to model resolution and the many Finish islands. But are these of interest to this study or can they be omitted from the presentation for (this readers) improved readability?

*In the figures based on the GCM ensemble the cover the same area as the Finnish gridded climate data set as that data set was used in the bias-correction. In the figures based on the RCM ensemble, the data coverage is different and data are missing from some coastal inland areas as well.*

Shoop, S. A., Vehicle bearing capacity of frozen ground over a soft substrate. Canadian Geotechnical Journal, 1995, Vol.32, No.3 :pp. 552-556. https://doi.org/10.1139/t95-057(http://www.nrcresearchpress.com/doi/abs/10.1139/t95-057#.Wr9Lki5uapo)

*We can add this reference.*

---

## Author Response (AR1)

In the beginning of this file, we provide our point-by-point responses to the reviewers' comments discussing any relevant changes made to the manuscript. A marked-up version of the revised manuscript follows then starting from page 10.

**Journal:** Hydrology and Earth System Sciences
**Title:** Projected decrease in wintertime bearing capacity on different forest and soil types in Finland under a warming climate
10 **Authors:** I. Lehtonen, A. Venäläinen, M. Kämäräinen, A. Asikainen, J. Laitila, P. Anttila and H. Peltola
**MS No.:** hess-2017-727
**MS Type:** Research Article
**Iteration:** First review
**Referee #1**

*We would like to thank the referee for the constructive comments and suggestions. Our replies to the comments describing the changes we have made into the manuscript are given in "Italics" after each specific comment.*

20 The paper presents the results of a study aimed to evaluate the projected decrease in the bearing capacity of Finnish soils in function of the changing climate during the 21st century. The paper appears well written and the results are interesting for the scientific community, even if related to the specific territory of Finland. The method is general and can be applied also in other nations in which the wood harvesting is economically important.

25 However, a lack in this paper is the detail related to the choices of parameters performed in the model used, the description of the pre-processing procedures (inclusive of the choices of the several parameters used in this study), the statistical comment about the values (especially those selected as a result of several simulations), and in general a too short description about the consequences and the limitations of these choices on the interpretation of the results. In my opinion, this part deserves a

30 deepening, because it could help to evaluate the results and also give more strength and robustness to the conclusions. This is the reason for which I do not think that this paper could be accepted in the present form, but requires some modifications that, in my view, can be intended as minor. The list of requirements can be understood better by looking at the specific comments here listed page by page.

35 − Introduction: in my opinion, a too large part of the introduction is dedicated to explain the industrial problems, while a too small part is dedicated to the scientific problem and the models used.

*We have shortened the introduction by compressing the first two paragraphs. As well, we have added text about challenges related to soil frost modelling.*

– Page 4 lines 4-28: the equation proposed to estimate soil temperature seems not consider the effects of soil moisture (unless thermal conductivity is kept variable, but since there are no measures of soil moisture it is hard to consider such variations). A comment on this consideration may be required.

*We have added a new sentence in the beginning of the chapter 2.1.1 (p. 4, l. 6) to state that "This assumption simplified the model considerably with expense of its validity under extremely wet and dry conditions."*

– Page 5 line 15: regarding KT values, is the interval of values used significant for the considered soils?

*We have added a mention that compared to the study by Jungqvist et al. (2014), the upper limit for $K_T$ in the parameter optimization was increased from $1\ W\ m^{-1}\ K^{-1}$ to $2\ W\ m^{-1}\ K^{-1}$ in order "to better represent the range of soil types and measurement depths considered in our study." (p. 5, l. 16)*

– Page 5 lines 18-19: "while, for example, KT seemed to steadily increase with soil depth." this is consistent with the assumption of increasing soil moisture at increasing depth (or change of soil texture): do you have any data evidencing these facts? Please comment.

*This is undoubtedly one reason for this. We have added a mention that the increase in $K_T$ with increasing depth is (p. 5, l. 20) "assumely largely due to increasing soil moisture with increasing depth (Soveri and Varjo, 1977)." In this report Soveri and Varjo (1977) showed in Table 4 measured soil moistures from one test site over one winter season. Based on those observations, the soil moisture, on average, increases with increasing soil depth. In Table 5 they showed typical heat capacity and heat conductivity values for different soil types with different soil moistures. Based on those values, the heat conductivity increases rapidly with soil moisture. In the report by Heikinheimo and Fougstedt (1992) are shown the soil textures at some of the stations on certain depths. For example, at Anjala the share of clay increases with increasing depth whereas, e.g., at Maaninka there is almost equal amount of silt and sand near the surface and also at 0.7 m depth but mainly sand around 0.5 m depth and below 0.7 m.*

– Pages 4-5: the method elaborated to retrieve soil thermal conductivity is strongly linked to the availability of soil temperature data, and thus will become representative of the experimental sites during the measurement periods. If I have correctly understood, such values optimized for each site will be adopted for the following simulations. However, there is no any reason for which such values could remain constant also in future climate... This could be a limitation for the reliability of future simulations. If authors do not agree with my conclusion, they could explain why...

*We have extended the discussion in the Section 4.1 considering many uncertainties in this study in general. Note also that for $K_T$ which seemed to be the most sensitive parameter, we finally used only the values optimized for one station representing each soil type and the rest of stations were rather used as validation stations. This was partly because not all stations clearly represented one soil type but they were rather mixed and varying depending on the depth. As can be seen from Table S2, high $R^2$ values*

*were achieved also at many other stations for different soil types, e.g., at Maaninka both for clay/silt and sandy soil. The model performed reasonably well even with the wrong soil type at many locations during the calibration period (see Table S2). As the used stations are moreover located in areas representing quite different climatic conditions, we assume that possible changes in soil characteristics, including thermal conductivity, do not crucially change the results.*

− Page 6 lines 1-10: the choice of different thresholds for soil freezing changed substantially the evaluation of the number of days with frozen soil? How and how much?

*We have elaborated that introducing the below-zero freezing points reduced the number of soil frost days at the depth of 20 cm (p. 6, l. 13-14) "only by a few days in sand but roughly by one month in clay/silt and approximately by 1–3 months in peat."*

− Page 6 lines 17-23, and page 7 line 5: I suggest to say here that the values used in eqs. 5 and 6 will be discussed later.

*We have added a notation that "the parameter values used in snow calculations will be discussed in the next chapter" (p. 7, l. 14)*

− Page 8 lines 3-12: again, the method elaborated to retrieve the values of parameters is strongly linked to the availability of measured data, and thus will become representative of the experimental sites during the measurement periods. Since such values optimized for each site will be adopted for the following simulations, there is no any reason for which such values could remain constant also in future climate... Also in this case, if authors have a different idea, they could explain why...

*The parameters optimized at each station located in different climatic conditions across Finland were averaged over all the stations to achieve the final parameters, which were then used in validation of the snow model (except kmax and kmin related to the solar azimuth angle having the latitudinal dependence). The validation period moreover had different kind of winters, cold and mild, snowy etc. If we had used the parameter values optimized for each station, the $R^2$-values would have been approximately 0.01 higher (p. 8, l. 24). Based on Table S3, the snow model performed equally well in different climatic conditions in different parts of Finland. However, we clearly see that the model with the optimized parameters performed worse before 1981 than thereafter, which we think is largely attributed to the correction factor for solid precipitation (cps), which we think had been higher previously due to a larger measurement error. Of course, it is not impossible that there have been some shifting in other parameters as well. On the other hand, part of the parameters are linked to things like freezing point of water or solar azimuth angle, which we can easily assume to stay constant.*

− Pages 8-9 lines 29-4: in this paper, many decisions about parameters are just summarized by "hiding" the results. For instance, in this case, the choice of values for kmin and kmax is not justified, and the reader cannot understand how it has been made. In my opinion, this may deserve an additional subsection (similarly as all other choices of this model).

*The impact of forest canopy for kmin and kmax was estimated based on Vehviläinen (1992). We have elaborated the choice of these parameters in more detail by extending the discussion in the last paragraph of the Section 2.2.2. We have moreover emphasized in the Section 4.1 that (p. 13, l. 30 forward) "As we did not have any snow depth measurements from forested sites, the modelled snow depths for forested areas were susceptible for biases" but that the "differences between different vegetation types were small" in the number of the days with good bearing capacity.*

– Page 9 lines 9-10: authors use only R2 as indicator of good simulations. However, if – just for example - I would have a simulation in which simulated snow depth has almost the same time trend of observations, but a value that is double, R2 will be close to one even if the relative error will be 200%... I suggest to use also bias or standard error as a criterion to validate simulations, and not only use correlation coefficient (and, by the way, it is better to use R and not R2).

*Due to this issue, in calibrating the snow model we minimized root mean square error (p. 8, l. 17) instead of maximizing $R^2$. Because of that, the modelled and observed snow depths were, on average, close to each other during the calibration period (Table S3). Both relative error and $R^2$ are shown in Table S3.*

– Page 9 lines 30-32: how the GCM and RCM have been chosen (I think you should mention here more clearly that the detailed list of model chosen is reported in Table S3), and why those models, among the whole EURO-CORDEX dataset?

*We have justified the model choice by stating (p. 10, l. 14-16): "The GCMs were chosen on the basis of their skill to simulate present-day average monthly temperature and precipitation climatology in northern Europe. From the EURO-CORDEX archive we chose the set of models with the largest number of simulations available with a uniform bias-adjustment approach." We moreover mention that the used GCMs are listed in Table S4 and RCMs in Table S5.*

– Page 10 line 20: is the modeled annual average number of days evaluated as the average of all GCMs and RCMs, respectively?

*We have clarified that in producing the left panel of Fig. 2 we have used observational weather data in the model calculations. The middle and right panel have been revised as suggested in your last comment so they depict now the multi-model mean difference between the model ensembles and calculations using the observational weather data. The text has been corrected accordingly.*

– Page 12 lines 6-15: how large is the difference among model ensembles (separately for RCM and GCM) in the three climatic periods? I think that also this information is important to statistically locate your results. Section 3,4 and Figure 6, in my opinion, are not informative, as they mention only the two models giving the maximum and the minimum values, and not the distribution. As climate cannot be described just by extremes, but needs a complete statistical information, for the same reason I think that

the standard deviation or some equivalent statistical parameter can be more informative about the dispersion of individual model calculations.

*To support Fig. 6, we have introduced a new figure (Fig. 7) to better illustrate the distribution among the model ensembles and also to compare the projected changes between different time periods and forcing scenarios.*

– Table 3: the numerical values given for each parameter have too many digits, most of them without any statistical meaning. Instead of giving a number with too many not significant digits, authors should give a number and an error associated with the experiments and comparisons, like a = value ± error

*We now show the parameter values in the table uniformly with five significant digits. The values of some parameters are well-known (like Napier's constant) but many of the parameters could be changed without any large change in the model performance if several parameters would be simultaneously adjusted appropriately. The parameter values are intrinsically not exact but they document the values used in this study. In the case of $K_T$, the optimized values for each individual station as well as those used for different soil types are shown in Fig. 1 to illustrate the uncertainty related to the fit. We have moreover added a new supplementary Table (Table S1) where we show the average values with their standard deviations for the optimized soil temperature model parameters after the first optimization round at two depths close to the surface. We believe that this helps to illustrate the uncertainty related to the parameters. At this point, we set the final values only for $f_S$ and $Z_l$ because we wanted to optimize the rest of values again because at some stations and some depths, the fixed values deviated quite a much from the optimized values. As $Z_l$ seemed to vary randomly across the stations and depths, we simply averaged it over all the station and depths to get the final value of 6.8 m (moreover, this parameter was virtually insignificant near the surface) but for $f_S$ we selected a value from the upper end of our sampling range because this parameter tend to get higher values when optimizing the number of days with soil frost instead of temperature.*

– Figure 2: since it is hard to appreciate differences among the three figures, given the quite large interval of variation of the number of days, it could be better to plot, for second and third column, the differences among GCM and observations, and RCM and observations, respectively (similarly to what you did for Fig, 3). Or maybe you can add such figures, if you want to keep the total number of days.

*The figure has been amended as suggested.*

**Journal:** Hydrology and Earth System Sciences
**Title:** Projected decrease in wintertime bearing capacity on different forest and soil types in Finland under a warming climate
**Authors:** I. Lehtonen, A. Venäläinen, M. Kämäräinen, A. Asikainen, J. Laitila, P. Anttila and H. Peltola
5 **MS No.:** hess-2017-727
**MS Type:** Research Article
**Iteration:** First review
**Referee #2**

10 *We would like to thank the referee for the comments and suggestions. Our replies to the comments describing the changes we have made into the manuscript are given in "Italics" after each specific comment.*

Wood harvesting is an important part of the Finnish economy. The current national strategy (2014-
15 2020) is also to increase the economic output from this industry, and over the next decades increasing demand is expected to put further pressure on the annual wood harvesting. This has traditionally been a winter activity when the ground is frozen and the mobility is greater and impacts on the natural areas are smaller. The paper aims to evaluate the projected decrease in the bearing capacity of Finnish soils, especially dried peatland, under two different projected warming scenarios for the 21st century. The
20 wood harvesting should preferably be undertaken (and increased) throughout the year, due to a steady demand by the processing industry. The results are presented across Finland for different combinations of three soil types and four forest types, in addition to forest truck roads. Hence, the results take into account that for a given location, the soil and/or forest might change in the future. The relevance of the study is therefore clear but the potential or expected impact could be better explained (in the context of
25 year-round harvesting with new methods and machinery in a changing climate) also because this study should have interest outside Finland and Finnish interests. The abstract provides a concise and complete summary, with the exception of not mentioning the snow model and using vague terms like "largely determined by" and "mainly determined by". Overall the paper is well structured and clear but could be shortened. The language is generally good with a few exceptions which will be fixed by a critical
30 shortening of the text.

*We thank the reviewer for these remarks. We have rephrased the mentioned expressions in the abstract and also included a mention about the snow model already in the abstract.*

35 The bearing capacity of frozen soil is an important parameter is this study. The authors claim, with reference to Eeronheimo (1991), that a 20 cm thick layer of frozen soil or 40 cm thick layer of ground snow can bear standard (heavy) vehicles (15-30 tons) used in forest harvesting. Eeronheimo is written in Finnish with an English summary at the end where it is stated: "The logging conditions in peatland forests are often unfavourable. The bearing capacity of the ground is poor most of the year and difficult
40 to determine, which complicates felling and extraction practices. According to forest harvesting specialists there should be either frost layer of at least 20 cm or, when there is no frost, snow cover of 40 cm or more to facilitate extraction with medium-sized forwarders." Using published guidelines,

Shoop (1995) proposed a relationship for bearing capacity of frozen ground, including as a function of dry and wet soil conditions. Dry conditions (dried peatlands are abundant in Finland) require a deeper freezing layer and a 10-ton truck needs 0.35-0.50 m frozen soil thickness, according to Table 1 in Shoop (1995), where the upper limit (0.50 m) is said to be a conservative estimate (page 555). (In the list of studies citing Shoop there are also more recent studies although they do not seem to bring any significant new knowledge on bearing capacity of frozen soil). In addition the distribution of the ground pressure (e.g. wheel load) from the vehicles/machines may impact the bearing capacity (breakthrough failure versus localized crushing). The relationship between the frozen soil layer and vehicle/machine bearing-capacity therefore seems more complex than put forward by the authors in this study. At least soil freezing depth and bearing capacity requires a more in-depth discussion (explaining also what is meant by "idealized approach" on page 13 line 8), but possibly a new analysis of the results. The claim that 40 cm thick ground snow (over frost free soil) ensures the same bearing capacity as 20 cm frozen soil should be supported by more evidence than from Eeronheimo. For instance, is this regardless of snow density? I guess not.

*We have extended the discussion related to uncertainties in our study and, e.g., to the chosen thresholds, in the Section 4.1. In addition, we have clarified that the requirement for 20 cm depth of soil frost or 40 cm depth of snow cover is based on experiences from forest harvesting specialists. Moreover, we have removed the word "idealized."*

It is correct that there are "several models designed for calculation of soil temperatures" (page 3 line 14) but the authors list only a few. For instance SURFEX (including different snow pack models like Crocus), see Special Issue in GMD at https://www.geosci-model-dev.net/special_issue14.html), where FMI has experience and expertise. There are most likely good reasons for the author's choice of land surface model but they are not well presented (and discussed) in the paper. Moreover, the employed GCMs and RCMs use land surface models and it should be explained why direct use of these results for soil temperature, or stand-alone high-resolution implementations, are not presented or used. This also applies to snow depth. I can speculate about the answer but not all readers might be well enough into surface modelling to do so. More details are also relevant for the validation of the modelled temperatures and snow depth. For soil temperatures the validation of the optimized model shows that either correlation or the number of days with soil temperatures below freezing can be well represented (but what is meant by "greatly overestimated" or "dramatically worse", top of page 6?). The results (correlation and number of freezing days) after introducing soil freezing points below zero are also not presented. These below zero freezing points are based on a study in Finnish and therefore difficult to use as reference – are there other studies which are more appropriate to use?

*We have extended the discussion in the Section 4.1 also by considering these aspects. Moreover, we have clarified the mentioned impressions in the top of page 6 (Sectio 2.1.3) and added information about the effect of modified freezing points on the number of days with frozen soil. Note that the correlation between modelled and observed soil temperatures was not affected by the artificial change in the freezing point. Finally, we have added two more references to studies dealing with below zero freezing points in the soils.*

Another reference should also be used for the determination of precipitation phase. Also, it should be explained why this classification is needed as not all readers are familiar with the (limited) output from numerical weather and climate simulations.

*We have added a reference also to Vehviläinen (1992).*

About the snow model, it is based largely on another study. From the text it is not very clear what is new and what is taken from the existing model. Can the text be shortened accordingly?

*Vehviläinen (1992) describes different approaches that can be used in snow cover modelling and we have took relevant pieces from these approaches. As Vehviläinen (1992) concentrates mainly on snow water equivalent (SWE), we have added also features describing the density of snow cover as Vehviläinen (1992) did not clearly describe how SWE should be transformed into snow depth. However, we have clarified in the beginning of the Section 2.2.1 that the used snow model is not completely similar to any existing model but based on the approaches described by Vehviläinen (1992).*

Is the modelled snow depths validated for RMSE? Or for the snow depth threshold of 40 cm?

*The snow model was optimized based on RMSE. $R^2$ and relative error are used to describe the validity of the model (see Table S3).*

The calibration periods for soil temperature and snow depth are relatively short. Is there a risk that they are too short when the resulting models are used on future climate conditions, cf. last sentence in Section 2.2.3?

*We assume that the relatively poor model performance before 1981 is related to higher undercatch of snowfall. We have added a reference to the study by Taskinen and Söderholm (2016) where they demonstrate that higher correction factor for precipitation is needed before 1981.*

The study "modelled the number of days with good bearing capacity in the forest harvesting point of view. (...) soil frost (...) at least the depth of 20 cm or when the snow depth exceeded 40 cm" (page 9, lines 27-29). I think it would add to the study to present the bearing capacity separately for soil frost thickness and snow depth. Both variables are dependent on climate change but their sensitivities to changes might be different, e.g. freezing temperature before or after snowfall affects the soil freezing layer differently. Also, such a separation should be beneficial both for the validity of the used model and when projected values are compared to current conditions (baseline period).

*We have introduced a new chapter (Section 3.5) including Fig. 8 dedicated to this issue.*

The evaluation of the methodology (Section 4.2) should be extended, following the suggestions above. Also, the comparison to Eeronheimo (and his Fig. 4) – page 13, lines 9-12) – could be done more

thoroughly, perhaps using a map of soil bearing capacity which reflects the actual soil and forest properties in each grid point.

*The evaluation of the methodology (Section 4.1) has been extended based on the suggestions above. The map of Eeronheimo indicates the soil bearing capacity only in peatland forests so it can be compared to our results for pine-dominated peatland forests.*

Are the GCM and RCM ensemble means used in Fig. 2?

*We have updated the figure and multi-model ensemble mean differences compared to calculations using the observational weather data are now shown in the middle and right panels.*

I might be wrong, but it seems that in several of the panels in Figs. 2-6 the bearing capacity also extends over sea. I am guessing this is related to model resolution and the many Finish islands. But are these of interest to this study or can they be omitted from the presentation for (this readers) improved readability?

*We have updated the figures following these suggestions.*

Shoop, S. A., Vehicle bearing capacity of frozen ground over a soft substrate. Canadian Geotechnical Journal, 1995, Vol.32, No.3 :pp. 552-556. https://doi.org/10.1139/t95-057(http://www.nrcresearchpress.com/doi/abs/10.1139/t95-057#.Wr9Lki5uapo)

*We have added this reference.*

[revised manuscript text omitted]
 3. In addition, to illustrate uncertainty in the parameter values, we show in Table S1 the optimized values with their standard deviations after the first optimization round averaged over all the validation stations at 10 cm and 20 cm depths.

**2.1.3 Validity of the modelled soil temperatures**

15  Apart from the stations used in calibration of $K_T$ for different soil types (Lettosuo, Anjala and Sodankylä), the modelled soil temperatures for clay/silt and sand soil types typically explained 90–99% of the observed variability in soil temperatures between the depths of 20 and 100 cm (Table S2+). Near the surface the modelled temperatures correlated slightly worse with the observed ones, as well as below 1 m. In spite of the generally high correlations, the modelled number of days with soil temperatures below 0 °C were still greatly overestimated (not shown). , even by more than twofold on many stations. We also

20  tested setting the model parameters by optimizing the modelled number of days with soil temperatures below 0 °C but then the correlations between observed and modelled soil temperatures became dramatically clearly worse. $R^2$ values dropping below 0.9 even at the best. In order to estimate more realistically the number of days with frozen soil, we thus assumed that the soil does not freeze completely until the soil temperature drops below –0.1 °C in sand or below –0.5 °C in other soil types as some supercooling in the soil is needed to initiate the process of freezing (Kozlowski, 2009). For instance, in kaolinite clay

25  ice lenses start to form in temperatures between –0.2 °C and –0.3 °C based on experiments and theoretical calculations (Style et al., 2011). At the depth of 20 cm, this reduced the number of soil frost days only by a few days in sand but roughly by one month in clay/silt and approximately by 1–3 months in peat. This The choice of freezing points was based on a study by Soveri and Varjo (1977) who stated that the freezing point in saturated sandy soil lies between 0 and –0.15 °C and in thin clay around –0.5 °C. Based on their study, in thick clay the freezing point can be as low as –20 °C, because the finer soil texture is, the

30  stronger absorption and capillary water bound around the soil particles by reducing the freezing point. The melting point of soil was still set to 0 °C in all of our calculations.

**2.2 Outlines for snow model and its parametrization and validity**

**2.2.1 Outlines for snow model**

In order to estimate snow depth $D_S$ needed in the soil temperature calculations, we used a temperature index snow model based largely on approaches presented by Vehviläinen (1992). Meteorological variables needed in the snow depth calculations are daily mean air temperature and daily total precipitation sum. First, the precipitation is divided into liquid and solid forms of precipitation as follows (Hankimo, 1976; Vehviläinen, 1992):

$$P_{solid} = P_{tot}, \text{ when } T_{mean} \leq -2.0 \text{ °C}$$

$$P_{solid} = \left(\frac{-T_{mean}}{8} + \frac{3}{4}\right) \cdot P_{tot}, \text{ when } -2.0 \text{ °C} < T_{mean} \leq 0.0 \text{ °C}$$

$$P_{solid} = \left(\frac{-25T_{mean}}{90} + \frac{3}{4}\right) \cdot P_{tot}, \text{ when } 0.0 \text{ °C} < T_{mean} \leq 0.9 \text{ °C}$$

$$P_{solid} = \left(\frac{-5T_{mean}}{8} + \frac{17}{16}\right) \cdot P_{tot}, \text{ when } 0.9 \text{ °C} < T_{mean} \leq 1.3 \text{ °C} \tag{5}$$

$$P_{solid} = \left(\frac{-T_{mean}}{8} + \frac{33}{80}\right) \cdot P_{tot}, \text{ when } 1.3 \text{ °C} < T_{mean} \leq 3.3 \text{ °C}$$

$$P_{solid} = 0, \text{ when } T_{mean} > 3.3 \text{ °C}$$

$$P_{liquid} = P_{tot} - P_{solid}$$

where $P_{solid}$ (mm) is the amount of solid precipitation, $P_{liquid}$ (mm) is the amount of liquid precipitation, $P_{tot}$ (mm) is the total amount of precipitation and $T_{mean}$ (°C) is the 2-metre daily mean air temperature.

The used snow model calculates the snow water equivalent (SWE) and density of snowpack. SWE (mm) is divided into two components as follows:

$$SWE = SWE_{new} + SWE_{old} \tag{6}$$

where $SWE_{new}$ (mm) is the amount of SWE accumulated on the day considered and $SWE_{old}$ (mm) describes the amount of snowpack left from the previous day. $SWE_{new}$ is calculated as follows:

$$SWE_{new} = \text{cps} \cdot P_{solid} + SWE_{inc,liq} \tag{7}$$

where cps is a correction factor for solid precipitation and $SWE_{inc,liq}$ (mm) is the increase of water storage in snowpack due to liquid precipitation. $SWE_{inc,liq}$ is limited by the water retention capacity of snowpack (WH) which is proportional to the total amount of snowpack and is thus determined as follows:

$$WH = a \cdot SWE_{old} \tag{8}$$

where $a$ is an empirical coefficient. The parameter values used in snow calculations will be discussed in the next chapter. $SWE_{inc,liq}$ is furthermore defined as follows:

$$SWE_{inc,liq} = P_{liquid}, \text{ when } P_{liquid} \leq WH$$

$$SWE_{inc,liq} = WH, \text{ when } P_{liquid} > WH \tag{9}$$

Decrease of SWE is caused both by evaporation from snowpack and by melting. Snowmelt is caused by thaw and liquid precipitation. Rainfall affects snowmelt directly by heating snowpack but more importantly, also by creating drains in the snowpack and accelerating the ripening process of snow cover. $SWE_{old}$ is then calculated as follows:

[revised manuscript text omitted]

---

## Author Response (AR2)

**Journal:** Hydrology and Earth System Sciences
**Title:** Projected decrease in wintertime bearing capacity on different forest and soil types in Finland under a warming climate
**Authors:** I. Lehtonen, A. Venäläinen, M. Kämäräinen, A. Asikainen, J. Laitila, P. Anttila and H. Peltola
**MS No.:** hess-2017-727
**MS Type:** Research Article
**Iteration:** Second review
**Referee #1**

*We would like to thank this referee for the positive feedback. Our replies to the comments are given in "Italics" after each specific comment.*

The manuscript has improved significantly since the first review and I recommend publication after some technical corrections:

1) The language can benefit from the help of a native speaker. The manuscript is well written and easily readable but here and there some improvements in the English language will benefit the presentation and overall impression.

*Copy-editing service is provided in this journal as a part of the publication process.*

2) Could consider to include a reference to damage to the forest terrain in the first sentence of the abstract, and to climate in the last sentence on page 1.

*We have added into the end of the abstract a mention that "This is also needed to avoid unnecessary harvesting damages, like rut formation on soils and damage to tree roots and stems."*

3) I agree with the approach in Section 2.1.3 wrt subzero freezing point but it is not entirely clear how this affects the comparison between observed and modeled days with frozen soil, cf. the title of the section - Validity of the modeled soil temperatures.

*We have added a mention that this assumption concerning subzero freezing points was used in model calculations only. From the observations, it was not so obvious either to estimate the number of days with frozen soil because from most of stations observations were available only every fifth day and the observed soil temperature was often 0.0 °C in winter. These cases were considered non-frozen. In the model, however, cooling of soil does not stop when temperature reaches 0 °C, although it remarkably slows down due to $C_{ICE}$ but this slowing of cooling does not start before soil temperature has already dropped below 0 °C. This is basically the reason why it is not meaningful to consider the soil being frozen right after the modelled soil temperature has dropped below 0 °C.*

**Journal:** Hydrology and Earth System Sciences
**Title:** Projected decrease in wintertime bearing capacity on different forest and soil types in Finland under a warming climate
**Authors:** I. Lehtonen, A. Venäläinen, M. Kämäräinen, A. Asikainen, J. Laitila, P. Anttila and H. Peltola
**MS No.:** hess-2017-727
**MS Type:** Research Article
**Iteration:** First review
**Referee #2**

*We would like to thank this referee for the in-depth review. Our replies to the comments describing the changes we have made into the manuscript are given in "Italics" after each specific comment.*

Overall assessment

I believe that the study is interesting and of potential interest to the readers of HESS. However, though the manuscript has improved during the revision process, there are still significant issues that need to be addressed. My main concern is the poor description of the methodology, which is of particular interest to the audience of this journal. I find the description of the methodology quite confusing and lacking important information as described in more detail below. Some aspects of the results and discussion are also not clearly explained (details also below), which is at times aggravated by poor English usage.

*We have improved the description of the methodology with the help of specific comments provided by the reviewer.*

Detailed comments linked to text

The authors refer to "model optimization" throughout the manuscript (starting with the abstract). They are indeed optimizing parameters but with the goal of "calibrating" the model. So this should be clear in the abstract and throughout the manuscript, because within our community model optimization often refers to a very different aim (optimizing water allocation, etc); while here the use of optimization is very clearly focussed on calibration.

*We have changed the vocabulary as suggested.*

Sections 2.1, 2.2.2, etc.: titles for these subsections: It would be clearer to use the term "description" rather than "outline".

*We have replaced "outlines" with "description" in the titles.*

Section 2.1.2: Unfortunately, the description in this section is vague and unclear.

Page 5, Line 9, include the soil types for "Lompolojänkkä and Kaamanen ".

*We have clarified that "fens" are minerotrophic peatlands.*

Page 5, line 10-12: "measurements needed in the calculations were not available from Lettosuo, Lompolojänkkä and Kaamanen stations", the next lines mention that nearby stations are used, but it is not clear what stations.

*We have clarified which stations were used.*

Page 5, Lines 13-30: "the optimal parameter values were set to each station and each available measurement depth." This sentence is unclear. Do you mean that you calibrated the parameters for each station at different depths?

*Yes. We state now that "the parameter values were calibrated for each station and at different measurement depths"*

What parameters were "calibrated"? It seems that you include the parameters used in the calibration in Table 2, but you should mention them in the text, with a justification of why are these parameters selected (are you calibrating all possible parameters?).

*Table 2 includes the calibrated parameters. They are listed also in the text on page 4, lines 24-25.*

You also need to justify the selected parameters ranges for calibration.

*The sampling ranges were adopted from Jungqvist et al. (2014) but for $K_T$ the upper limit was extended from 1 W m$^{-1}$ K$^{-1}$ to 2 W m$^{-1}$ K$^{-1}$ to better represent the range of soil types and measurement depths considered in our study (page 5, lines x-x)*

Are you using a Monte Carlo approach?

*We assume that the approach can be described as "Monte Carlo approach" as according to our knowledge, "Monte Carlo approach" refer to almost any approach using random guesses in searching the solution. In principle, the calibration approach was adopted from Jungqvist et al. (2014) where they describe the approach as "a Monte Carlo sampling technique", although they use different terminology as the procedure was for them "model optimization", not "calibration."*

Please include a description of the calibration approach, with more technical details, with a more rigorous technical language as this will be easier to follow. For example :

lines 10-20 state: "For some parameters optimized values varied rather randomly within the sampling range between different depths and locations while, for example, KT seemed to steadily increase with soil depth, assumedly largely due to increasing soil moisture with increasing depth"

The calibration procedure needs to be described with more rigour, it is unclear what the behaviour of each parameter was... "some parameters varied randomly" which parameters? Why?

Line 21: "Zl was set to 6.8 m and fS to 9.0 at each location and depth" why these values?

*6.8 m was the average for calibrated $Z_l$ values over all the stations and all the measurement depths. In practice, the effect of heat flow from $Z_l$ was negligible at the depths near the surface. Considering $f_S$, we noted that the calibrated values varied between 9 and 10 with soil depths below 50 cm except at two stations. With increasing soil depth, calibrated $f_S$ values tended to decrease, implying that the effect of snow cover in controlling soil temperature decreases with increasing soil depth, as expected because*

*the relative importance of heat flow from the surface compared to the heat flow from $Z_l$ decreases with increasing soil depth.*

Line 23: "after this second optimization round, all other parameters except KT were also set to their final values." What do you mean by "final values"?

*We mean that these parameters were kept constant afterwards.*

Lines 23-24: "KT,LOW and CS,LOW were given the same values at all depths and locations while CICE was set to depend on the soil type and CS the depth following asymmetrical sigmoid function." There is no rationale on why this was done.

*We have added explanations for these choices in the text.*

Line 25: It seems that in the end only Ks is calibrated?

*Only $K_T$ was sampled in the final phase while other parameters were kept constant. This was done because less important parameters could have very variable calibrated values at different stations and depths but in order to describe different soil types with fixed parameter values, we needed to have fixed values for each parameter. As $K_T$ was apparently the most sensitive parameter – i.e., whatever values were given to the rest of parameters, a relatively good model fit could be always achieved by tuning $K_T$ values appropriately – in the final phase we sampled only $K_T$. We have extended discussion related to this choice. We have moreover added to the Discussion section note that a model with almost equally good fit could be achieved with many different set of parameters "because there is no single best model parameter set but many model state descriptions can generate equally good calibration outputs (Beven, 2006; Jungqvist et al., 2014)."*

Line 26: "Anjala, Sodankylä and Lettosuo stations were selected to represent clay/silt, sand and peat soil types, respectively" At what point in the calibration was this decided? Why? Shouldn't this be decide at the beginning?

*This was decided mainly based on the results seen in Fig. 1, but before fitting the regression curves for the calibrated $K_T$ values. Basically, all the stations were used in the calibration before that phase but at these stations, the calibrated $K_T$ values at different depths followed nicely the logistic regression. Moreover, at some of the stations there were variability in the soil type with different depths so they were not equally well representative for a specific soil type (this was a possible reason for the less clear relationship between soil depth and calibrated $K_T$ values at these stations) and additionally, there were no missing measurement depths at these stations. The explanation has been added to the text.*

Lines 27-34: The rest of this section is also unclear.

*We have regrouped the rest of section. In the revised version, we only explain how soil frost conditions on forest truck roads were calculated.*

Page 6, Section 2.1.3. Please clarify the procedure used here. The authors state:

Lines 2-5: "Apart from the stations used in calibration of KT for different soil types (Lettosuo, Anjala and Sodankylä), the modelled soil temperatures for clay/silt and sand soil types typically explained 90–99% of the observed variability in soil temperatures between the depths of 20 and 100 cm (Table S2)."

This sentence is unclear. You mention "apart from the stations" this suggests that you are going to use data from "other stations"?

*Yes, this refers to all other stations in the Table S2 than those three stations mentioned in the brackets. We have rephrased the sentence to make this clearer for the readers.*

Lines 5-6: "In spite of the generally high correlations, the modelled number of days with soil temperatures below 0 °C were still greatly overestimated, even by more than twofold on many stations."

Where are you showing this? Please refer to the table/figure explicitly.

*This is not shown with numbers in any table or figure. We have clarified that this is a not shown result.*

Page 7. Line 14. You are probably referring to the next section not chapter.

*We are not either aware where this sentence is referring to, so we have removed the whole sentence. A reviewer asked us to include this sentence during the previous peer-review round.*

Page 10. Lines 11-14 states: "The calculations for the period 1980–2099 under changing climate were completed using daily data from six GCMs (listed in Table S4) participating in the CMIP5 (Taylor et al., 2012; Flato et al., 2013), as bias-corrected and downscaled onto the Finnish grid". Please clarify: Table S4 does not contain much information. What are the variables/data that are being used? What is the "bias-correction" (note that you mention this later in some figures without any explanation. What is the downscaling? Please add a brief explanation, and if needed more info.

*As most of these issues were discussed in the last paragraph of this section, we have reorganized the section and extended this discussion. We have moreover added a reference to Maraun and Widmann (2018) for those readers who would like to have a more in-depth look into the topic of downscaling and bias-correction.*

Page 11. Section 3.1: Figure 2 is discussed in section 3.1. It presents the results of the model on number of days with satisfactory bearing capacity for 1981-2010, which are compared to observations. Lines 14-19 state: "the used models generally reproduce the spatial pattern of wintertime bearing season length during the baseline period as expected as the model data has been bias-correct". I am not sure what patterns are discussed here. The comparison of results from GCM/RCM to those using observation do not show a "general good agreement". I would argue that the agreement is very poor. The patterns based on observation show a gradual trend from southern to northern portions that is absent from the ones produced by the ensemble forecasts, these last ones have a more random pattern. Please clarify. Please also explain the potential impact of this lack of agreement on the projected trends for climate change.

*We have slightly modified the word choices in this section. However, we argue that the agreement between multi-model ensembles and observations is generally good as the number of days with good bearing capacity varies mainly between 60 and 210 days and the differences are in the case of GCM ensemble almost everywhere less than 5 days (with the except of pine forests on peatland) and also in the RCM ensemble only locally more than 10 days.*

Page 11. Section 3.2: Lines 25-26 is unclear. It is also not explained why the focus has already shifted to drained peatlands as opposed to the other soils. The discussion should focus on all soil types for this figure.

*We have modified the discussion related to these figures by removing the last sentence from the first paragraph because the differences in projected shortening of bearing season length are still small between the different forest types during the near-future period 2021–2050.*